# Rethinking the Evaluation of Out-of-Distribution Detection: A Sorites Paradox

**Xingming Long**[1,2], **Jie Zhang**[1,2*], **Shiguang Shan**[1,2], **Xilin Chen**[1,2]
[1]Key Laboratory of AI Safety of CAS, Institute of Computing Technology,
Chinese Academy of Sciences (CAS), Beijing, China
[2]University of Chinese Academy of Sciences, Beijing, China

## Abstract

Most existing out-of-distribution (OOD) detection benchmarks classify samples with novel labels as the OOD data. However, some marginal OOD samples actually have close semantic contents to the in-distribution (ID) sample, which makes determining the OOD sample a Sorites Paradox. In this paper, we construct a benchmark named Incremental Shift OOD (IS-OOD) to address the issue, in which we divide the test samples into subsets with different semantic and covariate shift degrees relative to the ID dataset. The data division is achieved through a shift measuring method based on our proposed Language Aligned Image feature Decomposition (LAID). Moreover, we construct a Synthetic Incremental Shift (Syn-IS) dataset that contains high-quality generated images with more diverse covariate contents to complement the IS-OOD benchmark. We evaluate current OOD detection methods on our benchmark and find several important insights: (1) The performance of most OOD detection methods significantly improves as the semantic shift increases; (2) Some methods like GradNorm may have different OOD detection mechanisms as they rely less on semantic shifts to make decisions; (3) Excessive covariate shifts in the image are also likely to be considered as OOD for some methods. Our code and data are released in https://github.com/qqwsad5/IS-OOD.

## 1 Introduction

Deep neural networks achieve excellent results in many areas like computer vision and natural language understanding. However, though these well-trained models perform well on in-distribution (ID) test data sampled from the same distribution with the training set, they tend to struggle when confronted with the data drawn from out-of-distribution (OOD). For example, encountering previously unseen classes can result in the model making overly confident predictions [1]. This highlights the importance of ensuring the model's safety on such OOD data. An important research focus is OOD detection, which aims to enable models to detect such OOD samples rather than making incorrect judgments about them. There has already been considerable research and notable progress in the field of OOD detection [1, 2, 3, 4, 5, 6, 7, 8]. Compared to works on OOD generalization that focus on models' robustness to covariate shifts (such as changes in data style), these OOD detection works are mainly dedicated to capturing the semantic shifts (such as the novel classes).

Simultaneously, various benchmarks are constructed to study the performance of the above OOD detection methods [9, 10, 11, 12, 13, 14, 15]. Most of these works use two datasets with non-overlapping semantic labels as the ID and OOD data to construct their benchmarks. The ID dataset is divided into two parts: the first part serves as the training set, and the second part is mixed with the

---

*Corresponding author

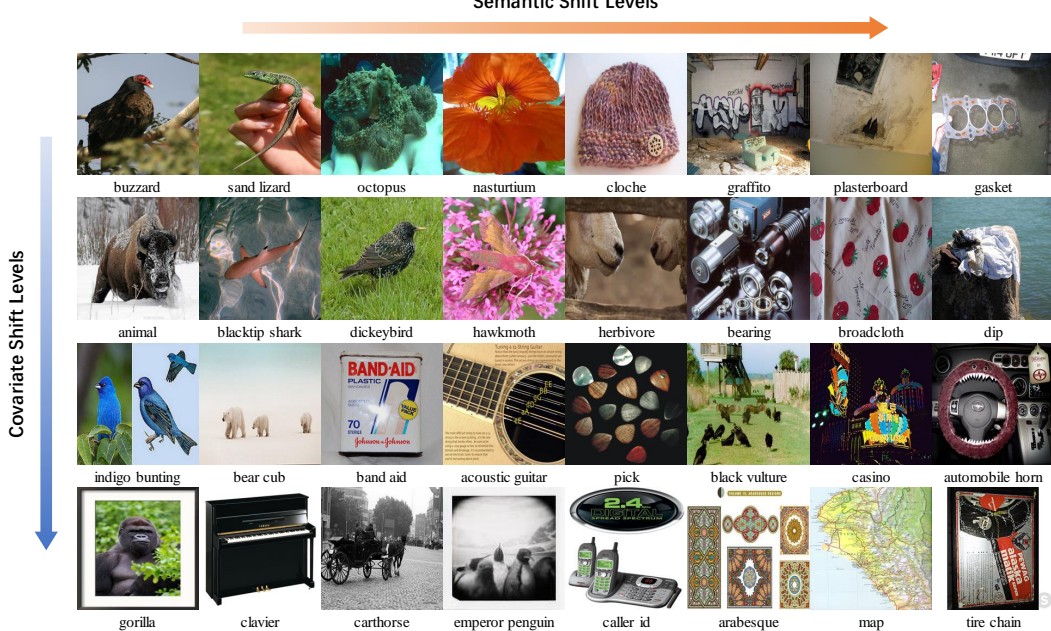

Figure 1: **Examples of images from IS-OOD benchmark.** ImageNet-21K is divided into subsets with different semantic and covariate shift levels relative to ImageNet-1K. As semantic shift increases, images of the subsets change from marginal samples (such as animal subspecies) to more distinct OOD categories (such as "gasket"). As covariate shift increases, the covariate contents transition from object-centered real photos to synthetic images, and from high-definition color images to low-resolution monochrome images.

OOD dataset to form the test set. The OOD detection model is trained on the training set and then evaluated on the mixed test set to assess its ability to detect the OOD data.

However, we find there exist drawbacks in the current OOD detection benchmarks that use semantic labels to distinguish OOD samples. Some marginal OOD test samples' semantic contents are actually very close to the ID data even if they have different semantic labels. The reason lies in the fact that the image's semantic labels are sometimes inaccurate; for example, they may omit background objects or common sense relationships portrayed in image [16]. This inaccuracy will lead to issues like similar labels (e.g., "African bush elephant" vs. "African elephant"), overlapping labels (e.g., "corn" vs. "food"), or insufficient labels (the label of the OOD sample do not contain the ID object in the image, e.g., an "athlete" wears a "T-shirt"), as illustrated in Figure 2. Although some researchers also recognize the problem and manually filter in-distribution data from the test sets [13, 14], these works are labor-intensive and may introduce subjective biases of individuals.

In our opinion, the key issue of "determining whether a data is an OOD sample" is actually a Sorites Paradox. Just like the dilemma of "how many grains of sand can be removed from a heap before it ceases to be a heap?", we cannot provide a precise definition for "how different a sample must be from the ID data to be considered an OOD sample?". Therefore, to address the issue, we need a method to measure "the degree of shifts relative to the ID data" ("how much sand has been removed") rather than continuing to debate "whether it is an OOD sample?" ("whether it ceases to be a heap?").

In this paper, we propose a shift measuring method and construct an Incremental Shift OOD (IS-OOD) detection benchmark that divides the test samples (from ImageNet-21K [17]) into subsets with different shift levels relative to the ID dataset (ImageNet-1K [18]), as shown in Figure 1. Considering the covariate content is also a potential influencing factor in OOD detection works [12], we propose a Language Aligned Image feature Decomposition (LAID) based on CLIP [19] features and thus measure the semantic and covariate shifts separately. Moreover, considering the limited covariate variation (such as limited types of styles) in ImageNet-21K, we construct a Synthetic Incremental Shift (Syn-IS) dataset that contains a series of high-quality generated images with more diverse

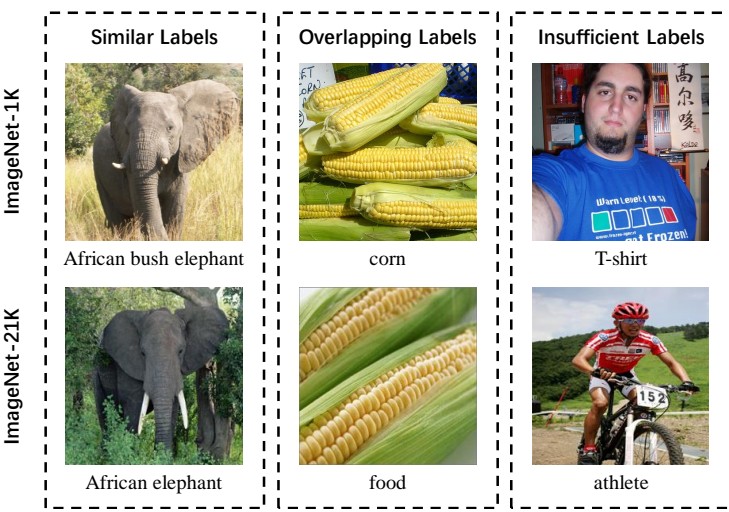

Figure 2: **Examples of noise caused by inaccurate semantic labels.** The images in the row below are semantically similar to the ID data (images in the row above), yet they are considered OOD samples in some benchmarks for their labels.

covariate contents to complement the IS-OOD benchmark. We discuss some existing benchmarks and compare them with our IS-OOD in Appendix A.

The main contributions of this work are summarized as follows:

- We construct an Incremental Shift OOD (IS-OOD) detection benchmark that divides the test samples into subsets with different levels of semantic and covariate shifts. We further generate a Synthetic Incremental Shift (Syn-IS) dataset that contains a series of high-quality images with more diverse covariate contents to complement the IS-OOD benchmark.

- We propose a Language Aligned Image feature Decomposition (LAID) method to obtain the semantic and covariate features of test images for shift measuring. Specifically, we utilize the decomposition of the CLIP's text features to determine the corresponding decomposition of the image features.

- We uncover several important insights with the proposed benchmark: (1) The performance of most OOD detection methods significantly improves as the semantic shift increases; (2) Some methods like GradNorm may have different OOD detection mechanisms as they rely less on semantic shifts to make decisions; (3) Excessive covariate shifts in the image are also likely to be considered as OOD for some methods.

## 2 Benchmark Construction

In this section, we will detail the construction of our benchmark. First, we will introduce our proposed Language Aligned Image feature Decomposition (LAID) method. Next, we will explain how we utilize the decomposition result to measure the shifts and construct the proposed Incremental Shift OOD (IS-OOD) benchmark. Following that, we will describe how we generate the Synthetic Incremental Shift (Syn-IS) dataset. Finally, we will briefly introduce the metrics we use for evaluating the OOD detection methods.

### 2.1 Feature Decomposition

We find that large-scale image datasets often lack covariate labels (such as the style or the augmentation of the image), and it is not easy to accurately decompose the covariate features based solely on the semantic labels. Although some datasets include covariate labels, such ImageNet-R [20] and PACS [21], they are small in scale and offer limited types of covariate labels.

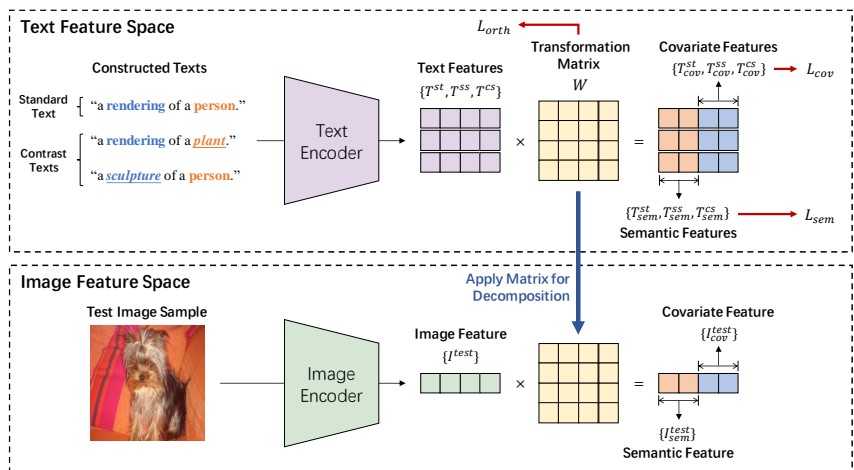

Figure 3: **Overview of Language Aligned Image feature Decomposition (LAID) method.** We first construct texts using different semantic and covariate prompts and train an orthogonal transformation matrix for the decomposition in the text feature space. Then, we can apply this matrix to the decomposition in the image feature space leveraging the alignment property of the CLIP model.

Inspired by many works that leverage the aligned text and image features of the CLIP model [22, 23, 24, 25], we propose the Language Aligned Image feature Decomposition (LAID). Given that text is easily editable, we can effortlessly construct a text dataset with diverse semantic and covariate contents through text concatenation. We then train a decomposition matrix in the text feature space and apply it to the image feature space using the alignment property of CLIP [19].

The overview of LAID is shown in Figure 3. To ensure the CLIP features' information is preserved, we use an orthogonal transformation matrix $W$ for the feature decomposition. After a feature $f \in \mathbb{R}^l$ undergoes transformation by the matrix $W \in \mathbb{R}^{l \times l}$, we designate the first half as the semantic feature $f_{sem} \in \mathbb{R}^{l/2}$ and the second half as the covariate feature $f_{cov} \in \mathbb{R}^{l/2}$:

$$\begin{aligned} f_{sem} &= (f \cdot W)[1 : l/2], \\ f_{cov} &= (f \cdot W)[l/2 + 1 : l], \end{aligned} \tag{1}$$

where $l$ represents the length of the CLIP feature.

The transformation matrix is optimized through contrastive learning. Specifically, in each training iteration, we construct a standard text and two contrast texts. One of the contrast texts exhibits only the semantic shift, while the other carries only the covariate shift. The semantic parts of these texts are selected from the ImageNet-21K labels. The covariate parts are derived from the prompts used in the CLIP zero-shot task [19], as these prompts contain a variety of covariate types like image styles and augmentation methods. The features of the standard text, semantic shift text, and covariate shift text, obtained by the CLIP text encoder, are denoted as $T^{st}$, $T^{ss}$, and $T^{cs}$. The relationships among these features are then constrained by triplet loss:

$$\begin{aligned} L_{sem} &= dist(T^{st}_{sem}, T^{cs}_{sem}) - dist(T^{st}_{sem}, T^{ss}_{sem}) + \alpha, \\ L_{cov} &= dist(T^{st}_{cov}, T^{ss}_{cov}) - dist(T^{st}_{cov}, T^{cs}_{cov}) + \alpha, \end{aligned} \tag{2}$$

where $dist(\cdot, \cdot)$ represents the cosine distance between two features, and $\alpha$ is a pre-defined margin.

Finally, we add a regularization loss to ensure $W$ remains an orthogonal matrix.

$$L_{orth} = \left\| W^T W - I \right\|_2^2. \tag{3}$$

After training on the constructed text dataset, we can achieve decomposition in the text feature space through the optimized transformation matrix $W$, thus obtaining the corresponding decomposition in the image feature space. Analyses regarding the effectiveness of the feature decomposition methods above are provided in Appendix B.

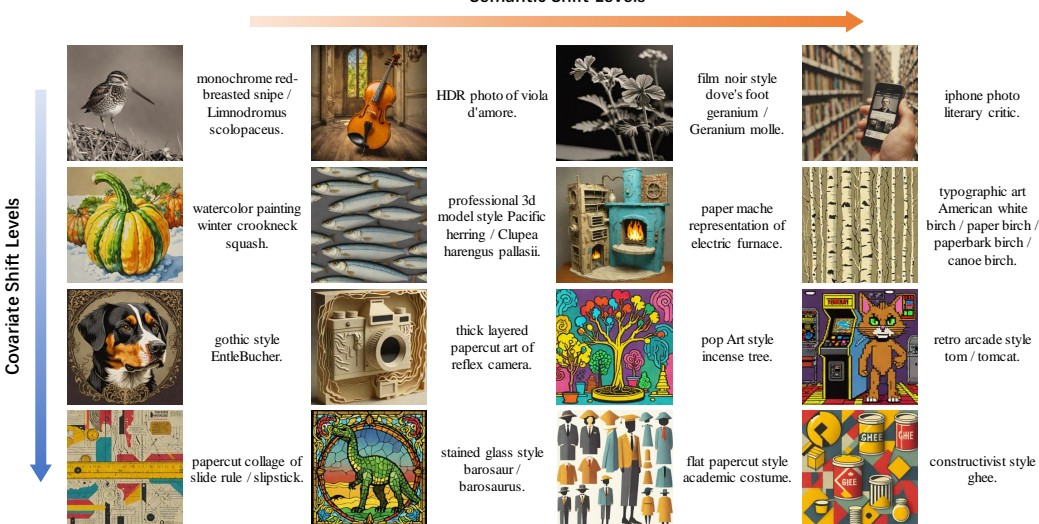

Figure 4: **Examples for the images in different Syn-IS subsets and their corresponding prompts.** Subsets with low covariate shifts typically include more realistic-style images (such as "HDR photo"), whereas subsets with high covariate shifts tend to contain more abstract-style images (such as "papercut").

## 2.2 Shift Measuring and Subsets Division

The measurement of the shifts between the test data (from ImageNet-21K [17]) and the ID dataset (ImageNet-1K [18]) is achieved based on the decomposition results above. We employ the decomposition method on the test sample and compute its semantic and covariate feature distance corresponding to each sample in the ID dataset. We can then use the nearest semantic or covariate distance between the test sample and the entire ID dataset to measure the degree of semantic or covariate shift for this test sample:

$$
\begin{aligned}
D_{sem}(I^{test}) &= \min_{I^{id}} dist(I^{test}_{sem}, I^{id}_{sem}), \\
D_{cov}(I^{test}) &= \min_{I^{id}} dist(I^{test}_{cov}, I^{id}_{cov}),
\end{aligned}
\tag{4}
$$

where $I^{test}$ and $I^{id}$ respectively represent the features of the test sample and the ID dataset.

We then categorize the test samples from ImageNet-21K into different shift levels according to their shift degrees. We split the shift degree in each of the semantic and covariate directions into 8 levels (a total of 8×8=64 subsets), ensuring a reasonable distribution while maintaining uniformity in the segmentation of intervals. ImageNet-21K dataset is thus divided into subsets with different semantic and covariate shift levels, and examples from different subsets are shown in Figure 1. Details of dividing the subsets can be found in Appendix C.

## 2.3 Generation of Syn-IS

We find that the covariate variation (such as the style types) is limited even in a large-scale dataset like ImageNet-21K. We also observe that as the quality of generated images improves, an increasing number of studies are using these generated images for visual tasks to address the weakness in existing datasets [26]. In order to enhance the diversity of the covariate components, we construct a Synthetic Incremental Shift (Syn-IS) dataset that contains a series of high-quality generated images with different semantic and covariate shift levels to complement our benchmark.

To ensure the covariate diversity of the generated image, we choose the official style templates provided by Stable Diffusion XL (SDXL) [27] as the covariate contents. The style templates are paired with ImageNet-21K labels to create the collection of prompts for the generation. The details of the prompts for generating Syn-IS are provided in Appendix D. We then divide the prompt collection into subsets with different semantic and covariate shift levels relative to ImageNet-1K. The degree of

shift is measured by the distance between the CLIP text features of these constructed prompts and the CLIP image features of ImageNet-1K. Each prompt subset is used to generate the corresponding subset of images with specific semantic and covariate shift levels.

We employ the SDXL-Turbo model (a distilled version of SDXL [27] based on Adversarial Diffusion Distillation [28]) to achieve the text-to-image generation. Examples of the generated images and their corresponding prompts are shown in Figure 4. It can be seen that the generated images indeed exhibit more diverse covariate contents, which effectively fills the gaps in ImageNet-21K and serves as a good supplementary to our benchmark.

## 2.4 Metrics

Following OpenOOD [9], we use three metrics to evaluate the performance of OOD detection methods on our benchmark: FPR@95, AUROC, and AUPR.

In addition to the three metrics above, we introduce two more metrics to study the changes in the model's performance across different shift levels. We first use the Pearson correlation coefficient to evaluate the relationship between the model's performance and the shift levels:

$$correlation = \frac{\sum_{i=1}^{n}(x_i - \bar{x})(i - \frac{n+1}{2})}{\sqrt{\sum_{i=1}^{n}(x_i - \bar{x})^2 \sum_{i=1}^{n}(i - \frac{n+1}{2})^2}}, \tag{5}$$

where $i$ represents the levels of semantic or covariate shift in the subset (1 being the smallest, $n$ being the largest), and $x_i$ represents the model's performance (such as AUROC) on shift level $i$.

To further investigate the extent of changes in model performance, we define a model's sensitivity to the corresponding shifts as follows:

$$sensitivity = \left| \frac{\sum_{i=1}^{n}(x_i - \bar{x})(i - \frac{n+1}{2})}{\sum_{i=1}^{n}(i - \frac{n+1}{2})^2} \right|. \tag{6}$$

The metric reflects the change in the model's performance when increasing one corresponding shift level. We assume that a good OOD detection model should have a high correlation with semantic shifts, and its performance should not vary significantly with covariate shifts. Therefore, we believe that a higher semantic sensitivity and a lower covariate sensitivity indicate a better method.

## 3 Experiments

In this section, we present the experimental results and corresponding findings from the proposed benchmark. During the experiments, each time we choose one divided subset from IS-OOD as the OOD data and mix it with ImageNet-1K test set to evaluate the OOD detection models trained on ImageNet-1K training set. Since the training data are from ImageNet-1K, all the shift levels mentioned in the experiments are the shifts of the test samples relative to ImageNet-1K. All evaluated OOD detection methods are implemented using a ResNet-50 [29] classifier trained on ImageNet-1K. The details of the evaluated OOD detection methods are introduced in Appendix E. Due to space limitations, only part of the results based on the AUROC metric are presented in this section. Complete experimental results including the metrics FPR@95 and AUPR can be found in Appendix F.

### 3.1 Main Results on ImageNet-21K

We first present the performance of the OOD detection methods on ImageNet-21K subsets according to different levels of semantic and covariate shifts. Due to the number of data in some subsets being too small, we omit these subsets and mark their results as "N/A". The results for part of the OOD detection methods are shown in Figure 5. From the results, we can make the following important observation:

**OOD detection methods perform better when there is a large semantic shift and a small covariate shift.** In the results, the subsets where these methods perform best are those with large semantic shifts and small covariate shifts. This observation confirms that OOD detection methods are not only sensitive to semantic shifts but also disturbed by covariate shifts. However, all these

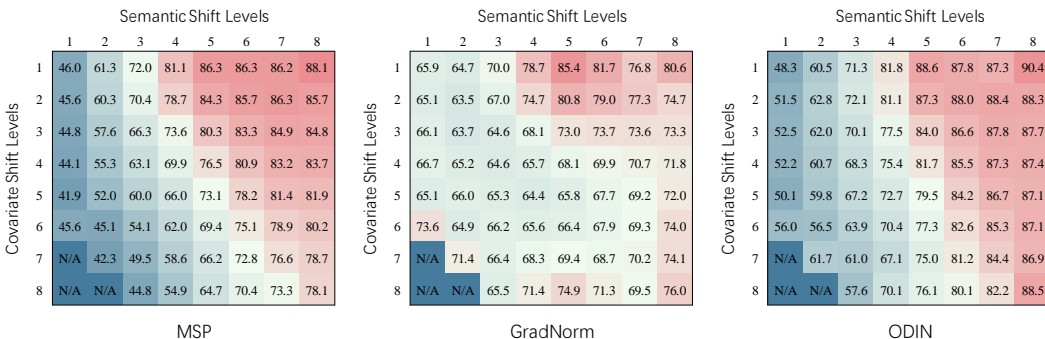

Figure 5: OOD detection performance on all ImageNet-21K subsets with different semantic and covariate shift levels. "N/A" indicates the number of data in this subset is too small for a fair evaluation.

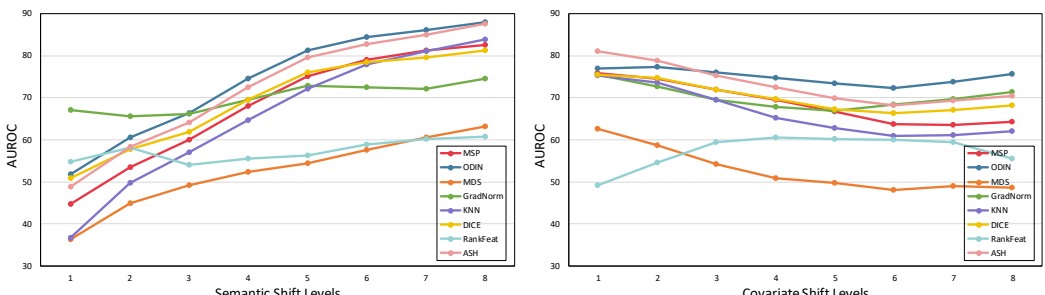

Figure 6: Comparison of OOD detection methods across different semantic or covariate shift levels on ImageNet-21K.

methods are not significantly affected by the covariate shifts. The primary factor influencing their performance is still the semantic shifts.

We then display how the performance of each method varies according to different levels of semantic shifts or covariate shifts separately, as shown in Figure 6. Specifically, we calculate the average result across different covariate shifts at each semantic shift level and draw the curves. The results for the covariate shift are obtained with the same approach. We can compare the performance of different methods and further derive the following insights:

**The performance of most methods significantly improves as the semantic shift increases.** It can be seen from the results for the semantic shift, that most OOD detection methods gain a nearly 40% increase in AUROC under the largest semantic shift compared to the result under the smallest semantic shift, except for GradNorm and RankFeat. Conversely, the AUROC of all the methods does not change a lot across different covariate shifts. The experimental results confirm the previous finding that the degree of semantic shifts is the primary factor that influences whether most OOD detection methods are capable of distinguishing between ID and OOD data.

**Some OOD detection methods rely less on semantic shifts to make decisions.** The AUROC of GradNorm and RankFeat do not significantly change with the semantic shift levels. We assume that these methods might make the prediction based on some low-level features that the CLIP encoders are unable to extract. This provides important insights for the study of the detection mechanisms of different OOD detection methods.

We calculate the correlation and sensitivity of each method, and the results are presented in Table 1. The results show that most methods exhibit positive correlation and higher sensitivity to semantic shifts while demonstrating negative correlation and lower sensitivity to covariate shifts, which is consistent with our discoveries above. Among these methods, KNN performs the best in terms of its semantic sensitivity. ODIN shows the lowest sensitivity to covariate shifts, meaning it is the least affected by changes in covariate contents. GradNorm and RankFeat obtain the lowest semantic sensitivity, which further suggests that these methods seem to rely less on semantic changes for OOD detection.

Table 1: Results of correlation and sensitivity for OOD detection methods on ImageNet-21K

| | Semantic | | Covariate | |
|---|---|---|---|---|
| | correlation | sensitivity | correlation | sensitivity |
| MSP [1] | 0.97 | 5.59 | -0.96 | 1.95 |
| ODIN [2] | 0.97 | 5.26 | -0.63 | 0.46 |
| MDS [3] | 0.98 | 3.50 | -0.91 | 1.98 |
| GradNorm [4] | 0.91 | 1.27 | -0.49 | 0.56 |
| KNN [5] | 0.98 | 6.64 | -0.93 | 2.18 |
| DICE [6] | 0.97 | 4.52 | -0.88 | 1.29 |
| RankFeat [7] | 0.78 | 0.79 | 0.51 | 0.83 |
| ASH [8] | 0.98 | 5.56 | -0.89 | 1.73 |

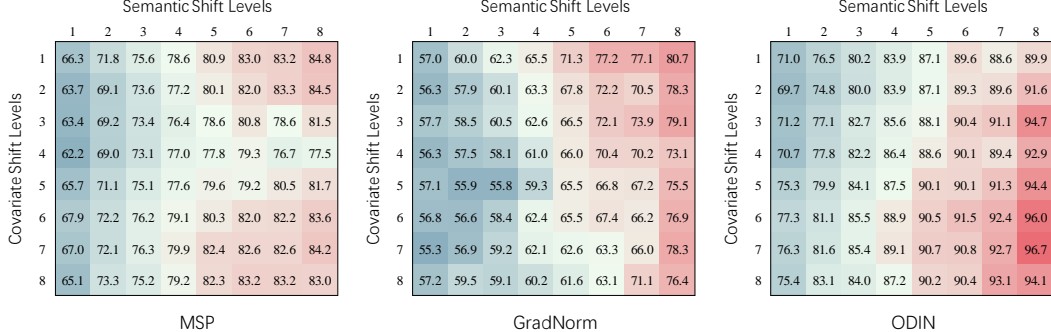

Figure 7: OOD detection performance on all Syn-IS subsets with different semantic and covariate shift levels.

## 3.2 Results on Syn-IS

Subsequently, we conduct experiments on the generated Syn-IS dataset using the same methods above, with the results shown in Figure 7, Figure 8, and Table 2.

From the results in Figure 8, we can see that the performance of most OOD detection methods still varies with semantic shifts, which further suggests that these methods are capable of conducting OOD detection based on the semantic contents even with generated data. However, we observe some interesting experimental insights that do not exist on the ImageNet-21K subsets.

**Samples with Excessive covariate shifts are also considered OOD.** We observe that, unlike the results on ImageNet-21K subsets where the covariate correlation of almost all methods is negative, many methods on Syn-IS show a positive correlation with the covariate shift levels. We assume this is because the covariate contents in ImageNet-21K are similar to those in ImageNet-1K, which means ImageNet-21K does not include samples with large covariate shifts (such as drastic style changes). In this case, the covariate shifts affect the extraction of the semantic features, thereby reducing the models' OOD detection capabilities. However, as the style shift continues to increase, such as the "papercut" and "constructivist" styles shown in Figure 4, the style information itself is so pronounced that it is seen as the OOD contents by the models, which then improves the models' OOD detection performance.

**The "generative" attribute of Syn-IS improves the performance of most models.** We observe that on Syn-IS, the semantic sensitivity of most methods decreases compared with the results on ImageNet-21K, as these models perform better at lower semantic shift levels on Syn-IS. We assume the reason is that generated images differ from real images. Even if the semantic and covariate contents of a generated image and a real image remain the same, the model can still distinguish them to some extent based on the unique patterns of the generated image.

**On Syn-IS dataset, the performance of GradNorm declines significantly.** We further analyze the principle of GradNorm and the characteristics of the Syn-IS dataset. We find that GradNorm tends to classify samples with uniform softmax outputs as OOD samples. Since each Syn-IS image is generated with a text that only includes one given label, the image is likely to contain a single object, unlike many images in ImageNet-21K that contain multiple objects. This difference could lead to less uniform softmax outputs on Syn-IS compared to Imagenet-21K, resulting in lower OOD

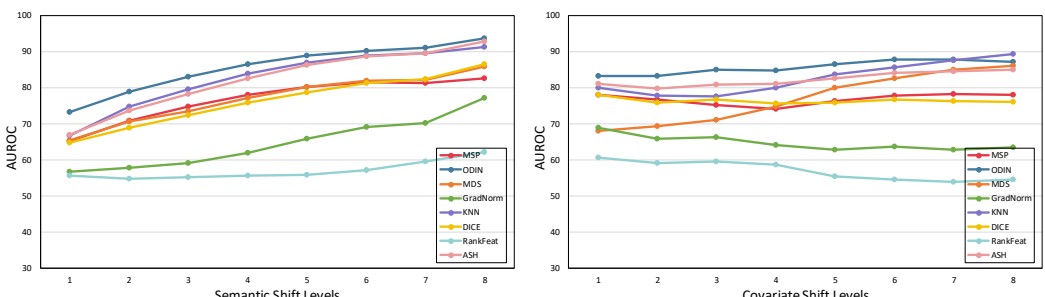

Figure 8: Comparison of OOD detection methods across different semantic or covariate shift levels on Syn-IS.

Table 2: Results of correlation and sensitivity for OOD detection methods on Syn-IS

|  | Semantic | | Covariate | |
|---|---|---|---|---|
|  | correlation | sensitivity | correlation | sensitivity |
| MSP [1] | 0.93 | 2.33 | 0.36 | 0.23 |
| ODIN [2] | 0.96 | 2.71 | 0.92 | 0.72 |
| MDS [3] | 0.98 | 2.71 | 0.99 | 2.91 |
| GradNorm [4] | 0.98 | 2.85 | -0.85 | 0.72 |
| KNN [5] | 0.95 | 3.30 | 0.92 | 1.71 |
| DICE [6] | 0.99 | 2.96 | -0.41 | 0.13 |
| RankFeat [7] | 0.87 | 0.91 | -0.94 | 1.03 |
| ASH [8] | 0.98 | 3.52 | 0.92 | 0.75 |

scores produced by GradNorm. This insight might highlight a potential limitation of the GradNorm approach.

## 4 Conclusion and Discussion

In this paper, we construct the IS-OOD benchmark that divides the test samples into subsets with different levels of semantic and covariate shifts relative to the ID dataset. Unlike most past works that rely on semantic labels, our benchmark utilizes the degree of shifts to categorize the test dataset, thereby avoiding the debate over determining whether a test sample is OOD. This benchmark helps in comprehensively analyzing the models' sensitivity to the semantic and covariate shifts. With our benchmark, we uncover several important insights: (1) The performance of most OOD detection methods significantly improves as the semantic shift increases; (2) Some methods like GradNorm may have different OOD detection mechanisms as they rely less on semantic shifts to make decisions; (3) Excessive covariate shifts in the image are also likely to be considered as OOD for some methods.

**Limitation:** The alignment between CLIP's text feature space and image feature space is not perfect, which may lead to a certain gap between the decomposition matrices in the two feature spaces. Future works can focus on narrowing this gap with a better vision-language model to enhance the accuracy of the shift measuring method in the benchmark.

**Societal Impact:** We conduct safety checks on both the prompts and the generated images for the Syn-IS dataset, which helps avoid potential negative societal impacts. As for our benchmark, it evaluates OOD detection models' sensitivity to test samples with varying shift degrees relative to the ID data, which is crucial for the safe deployment of the models. Therefore, we believe our work has positive societal impacts.

## Acknowledgment

This work is partially supported by Natural Science Foundation of China (No. U2336213), Strategic Priority Research Program of the Chinese Academy of Sciences (No. XDB0680202), Beijing Nova Program (20230484368), Suzhou Frontier Technology Research Project (No. SYG202325), and Youth Innovation Promotion Association CAS.

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

# A  Comparison with Other Benchmarks

The advantage of IS-OOD is the division of test samples into subsets with incremental levels of semantic and covariate shifts, which avoids the debate over "whether a test sample is OOD?" and allows for a more detailed analysis of the sensitivity of OOD detection models to the changes in semantic and covariate shifts. Compared to previous works that remove the noisy marginal ID samples from the OOD dataset [13, 14, 15], our approach does not involve manual annotation and thus introduces less subjective bias. Compared to a work that considers the impact of covariate contents [12], IS-OOD analyzes the covariate shifts in more detailed levels and considers them in conjunction with semantic shifts, which allows for a more comprehensive evaluation of how covariate contents affect the OOD detection model.

An interesting work also proposes to divide test data into different subsets for evaluation[11], in which ImageNet-21K is divided based on the OOD detection difficulty. However, they do not consider the impact of the covariate shifts in the benchmark. Besides, since the "detection difficulty" is derived from models trained only on the training set, its accuracy on the test data cannot be guaranteed. In our IS-OOD benchmark, we assume that the CLIP model is trained on a sufficiently large scale of data and is exposed to all the data within our benchmark, which ensures that both the text and image features from the CLIP model are accurate not only on our ID data (ImageNet-1K) but also on the test data (ImageNet-21K). Therefore, in the proposed benchmark, the CLIP model can be considered a "referee" capable of fairly determining the degree of semantic and covariate shifts for each test data.

# B  Analysis of the Proposed Decomposition Method

To analyze the effectiveness of the proposed Language Aligned Image feature Decomposition (LAID), we conduct zero-shot classification experiments using the decomposed feature components. Specifically, we first decompose the text features of the 1000 ImageNet-1K classes to obtain their corresponding semantic and covariate features. Then, we calculate the cosine similarity between the test images' semantic or covariate features and the above decomposed text feature components, in which way we achieve the zero-shot classification in the decomposed feature space. The results are shown in Table 3.

Table 3: Zero-shot results obtained with the decomposed features

|  | ImageNet-1K | | ImageNet-C | | ImageNet-R | |
|---|---|---|---|---|---|---|
|  | Acc@1 | Acc@5 | Acc@1 | Acc@5 | Acc@1 | Acc@5 |
| CLIP | **70.11** | **91.74** | **44.39** | **68.79** | 64.82 | 85.50 |
| Semantic Feature | | | | | | |
| PCA | 39.23 | 66.89 | 19.70 | 39.48 | 13.36 | 35.96 |
| LAID | 67.49 | 90.59 | 43.06 | 67.61 | **66.81** | **85.81** |
| Covariate Feature | | | | | | |
| PCA | 0.93 | 3.43 | 0.45 | 1.86 | 0.11 | 0.60 |
| LAID | 7.31 | 15.75 | 2.62 | 6.80 | 0.81 | 2.61 |

In the experiment, the test set of ImageNet-1K is used to evaluate the performance of ImageNet-1K. ImageNet-C contains ImageNet-1K data with various augmentations, and ImageNet-R includes ImageNet-1K classes in different styles. Regarding comparative methods, "CLIP" refers to the zero-shot classification results obtained using the complete CLIP features. Since our method is inspired by the Principal Components Analysis (PCA) method and also uses a transformation matrix to achieve feature decomposition, we compare the features obtained by the PCA method with those from our approach. Using the text dataset we constructed in Section 2.1, we perform PCA decomposition in the semantic and covariate directions separately. We select the dimensions accounting for the top 90% of cumulative variance as the decomposed feature for the "PCA" method.

The results show that using PCA as the decomposition method, the accuracy of semantic features drops significantly compared to the "CLIP", indicating that PCA decomposed features lost considerable semantic information compared to the complete CLIP features. In contrast, the proposed LAID method shows comparable performance with the "CLIP" and even shows improved accuracy on ImageNet-R, a dataset that contains obvious style shifts compared with ImageNet-1K. This suggests that the obtained semantic features successfully retain the complete semantic information and

decouple the covariate contents like the styles. Meanwhile, the accuracy using the covariate features is extremely low, indicating that the covariate features contain very little semantic information.

## C  Details of the Subsets Division

The process of dividing ImageNet-21K subsets based on the transformation matrix trained using LAID is detailed in Algorithm 1:

---

**Algorithm 1** Dividing the testing dataset.

---

**Input:** ID dataset $S^{id}$, testing dataset $S^{ood}$, CLIP image encoder $E$, semantic interval set $\text{Inter}_{sem}$, covariate interval set $\text{Inter}_{cov}$, transformation matrix $W$.
**Output:** Divided testing subsets $Out$.
1: Initialize the ID semantic and covariate feature sets: $C_{sem}$ and $C_{cov}$.
2: **for** $X^{id} \in S^{id}$ **do**
3:    Extract the ID image feature: $I^{id} = E(X^{id})$.
4:    Decompose the feature into semantic and covariate components:
         $I_{sem}^{id} = (I^{id} \cdot W)[1 : l/2], I_{cov}^{id} = (I^{id} \cdot W)[l/2 + 1 : l]$.
5:    Add features to the sets:
         $C_{sem}.append(I_{sem}^{id}), C_{cov}.append(I_{cov}^{id})$.
6: Initialize the output subsets: $Out$.
7: **for** $X^{test} \in S^{ood}$ **do**
8:    Extract the OOD image feature: $I^{test} = E(X^{test})$.
9:    Decompose the feature into semantic and covariate components:
         $I_{sem}^{test} = (I^{test} \cdot W)[1 : l/2], I_{cov}^{test} = (I^{test} \cdot W)[l/2 + 1 : l]$.
10:    Measure the shift degrees using the distance according to the nearest neighbor in the ID dataset:
         $D_{sem} = \min \left\{ dist(I_{sem}^{test}, I_{sem}^{id}), I_{sem}^{id} \in C_{sem} \right\}$,
         $D_{cov} = \min \left\{ dist(I_{cov}^{test}, I_{cov}^{id}), I_{cov}^{id} \in C_{cov} \right\}$.
11:    **for** $[start_i, end_i) \in \text{Inter}_{sem}$ **do**
12:      **if** $D_{sem} \in [start_i, end_i)$ **then**
13:         Determine the semantic shift level: $level_{sem} = i$.
14:    **for** $[start_j, end_j) \in \text{Inter}_{cov}$ **do**
15:      **if** $D_{cov} \in [start_j, end_j)$ **then**
16:         Determine the covariate shift level: $level_{cov} = j$.
17:    Add the OOD sample to the corresponding subset:
         $Out[level_{sem}][level_{cov}].append(X^{test})$.
18: **return** $Out$.

---

To avoid the noise that may result from using the nearest neighbor, we choose the $10^{th}$ nearest neighbor instead in our practical implementation to measure the shift degrees. The semantic and covariate interval sets are chosen to ensure a reasonable distribution while maintaining uniformity in the segmentation of intervals. The number of images in each subset after division is shown in Figure 9.

As observed in Figure 9, subsets with high semantic shifts and low covariate shifts (or vice versa) contain few images, sometimes even none. This indicates that there is a certain correlation between semantic and covariate information in ImageNet-21K. Therefore, we have to sacrifice the results on some subsets (marking the results of subsets with too little data as "N/A"), in which way the levels of shift can cover a broader range of shift degrees (a greater shift degree in the highest shift level). Notably, the correlation between semantic and covariate information in ImageNet-21K could potentially lead to incorrect evaluations of OOD detection models that are sensitive to covariate shifts. By creating a more balanced dataset like Syn-IS, the bias caused by this correlation can be mitigated.

(As for the Syn-IS dataset, each subset includes 5,000 generated images.)

Semantic Shift Levels

| | | 1 | 2 | 3 | 4 | 5 | 6 | 7 | 8 | Sum |
|---|---|---|---|---|---|---|---|---|---|---|
| | 1 | 472,201 | 251,593 | 153,321 | 103,532 | 56,383 | 11,354 | 3,156 | 898 | 1,052,438 |
| | 2 | 437,507 | 676,972 | 776,496 | 783,059 | 736,514 | 290,441 | 72,395 | 14,207 | 3,787,591 |
| | 3 | 83,146 | 268,596 | 518,514 | 687,373 | 840,241 | 669,000 | 304,351 | 90,249 | 3,461,470 |
| Covariate Shift Levels | 4 | 8,397 | 45,938 | 148,383 | 291,404 | 423,955 | 483,394 | 365,619 | 217,905 | 1,984,995 |
| | 5 | 661 | 5,146 | 23,035 | 63,651 | 118,179 | 178,828 | 198,955 | 246,555 | 835,010 |
| | 6 | 54 | 556 | 3,046 | 9,751 | 22,754 | 43,002 | 63,234 | 172,107 | 314,504 |
| | 7 | 9 | 56 | 335 | 1,357 | 3,548 | 7,998 | 13,988 | 75,631 | 102,922 |
| | 8 | 0 | 4 | 29 | 174 | 581 | 1,342 | 2,960 | 38,703 | 43,793 |
| | Sum | 1,001,975 | 1,248,861 | 1,623,159 | 1,940,301 | 2,202,155 | 1,685,359 | 1,024,658 | 856,255 | 11,582,723 |

Figure 9: The number of images in each ImageNet-21K subsets.

# D   Prompts Used for Syn-IS Generation

We filter out the styles that significantly distort semantic information or have dangerous tendencies, leaving a total of 51 different style templates for generating the Syn-IS. The names of these 51 styles are listed as follows:

"art nouveau", "constructivist", "expressionist", "graffiti", "hyperrealism", "impressionist", "pointillism", "pop art", "renaissance", "surrealist", "typography", "watercolor", "cybernetic", "cyberpunk game", "gta", "retro arcade", "retro game", "rpg fantasy game", "pixel art", "line art", "comic book", "3d-model", "dreamscape", "dystopian", "fairy tale", "gothic", "grunge", "kawaii", "lovecraftian", "macabre", "manga", "metropolis", "minimalist", "monochrome", "nautical", "space", "stained glass", "collage", "flat papercut", "paper mache", "paper quilling", "papercut collage", "papercut shadow box", "stacked papercut", "thick layered papercut", "film noir", "glamour", "hdr", "iphone photographic", "long exposure", "tilt-shift"

Each style contains a prompt and a negative prompt. An example is as follows:

**name:** "art nouveau"
**prompt:** "art nouveau style {object}. elegant, decorative, curvilinear forms, nature-inspired, ornate, detailed"
**negative prompt:** "ugly, deformed, noisy, blurry, low contrast, realism, photorealistic, modernist, minimalist"

The content of the prompt and the negative prompt for other style templates can be found at the SDXL website.

# E   Evaluated OOD Detection Methods

We test a total of eight typical OOD detection methods. These methods determine whether an input sample is OOD based on the model's output logits, features, and gradient statistics. Below, we will introduce these methods by category.

The importance of OOD detection is first proposed in [1], and **MSP** is introduced as the most basic baseline method for the task. **MSP** [1] uses the maximum softmax logits of the model's output to determine whether the input is OOD. If the maximum logit is high, it indicates that the model is confident about the input, thereby suggesting an ID sample. Conversely, if the maximum logit is low, the input is considered OOD. The subsequent works continue to use the logits for OOD detection and make several improvements. **ODIN** [2] introduces the temperature scaling and the input perturbations to improve the performance. **DICE** [6] ranks the weights of the last fully connected layer based on their contribution to the outputs and selectively uses the most salient weights to derive the logits for OOD detection. **RankFeat** [7] removes the rank-1 matrix composed of the largest singular value and the associated singular vectors from different levels of features. **ASH** [8] simply removes a large portion of a sample's activation before calculating the logits.

There are also feature-based methods. The main idea is that the greater the feature distance between the test sample and the ID training dataset, the more likely it is that the sample is OOD. **MDS** [3] computes the class-conditional statistics of the training features and detects the OOD samples based on the maximum Mahalanobis distance between the test sample and each class. **KNN** [5] performs OOD detection by directly measuring the L2 distance between the test sample and its K-nearest neighbors in the entire training set.

In addition to logits and features, gradients are also used for OOD detection. **GradNorm** [4] calculates the gradients of the parameters with respect to the entropy, which is then used as the confidence in determining whether an input is OOD.

# F Complete Experimental Results

## F.1 Complete AUROC Results on All Subsets

This section presents the AUROC of all the evaluated OOD detection methods on the ImageNet-21K subsets and the Syn-IS subsets, as shown in Figure 10 and Figure 11.

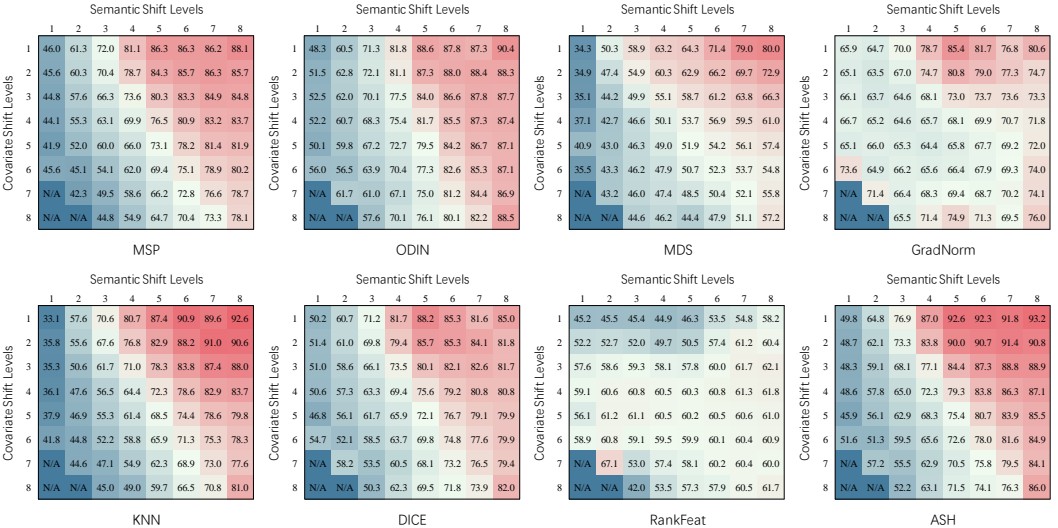

Figure 10: Methods' AUROC on all ImageNet-21K subsets with different semantic and covariate shift levels. "N/A" indicates the number of data in this subset is too small for a fair evaluation.

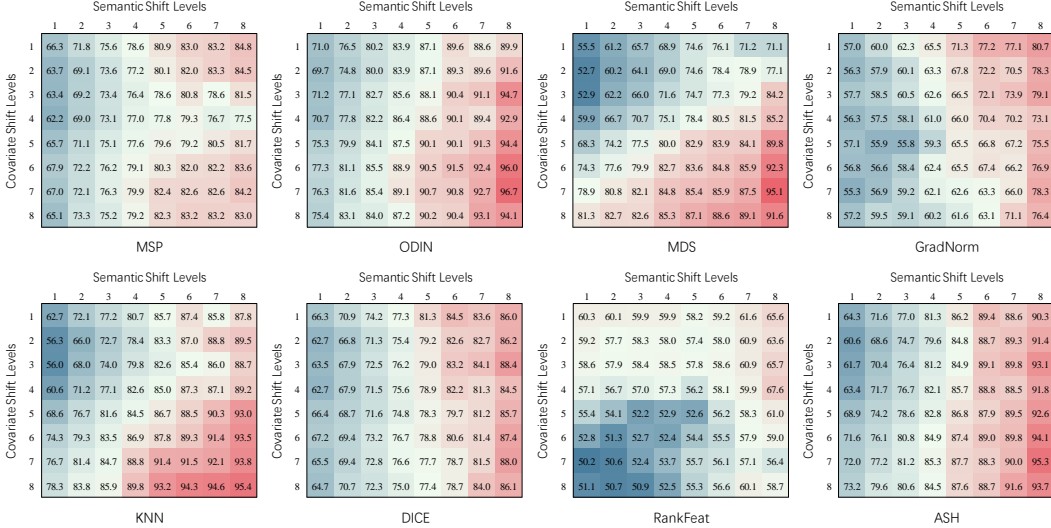

Figure 11: Methods' AUROC on all Syn-IS subsets with different semantic and covariate shift levels.

## F.2 FPR@95 Results on ImageNet-21K

This section presents the FPR@95 results of all the evaluated OOD detection methods on ImageNet-21K, as shown in Figure 12, Figure 13, and Table 4. Correlation and sensitivity in Table 4 are the two metrics we proposed in Section 2.4 used to study the changes in the model's performance across different shift levels. The performances of the methods are generally consistent with the AUROC results presented in the main text.

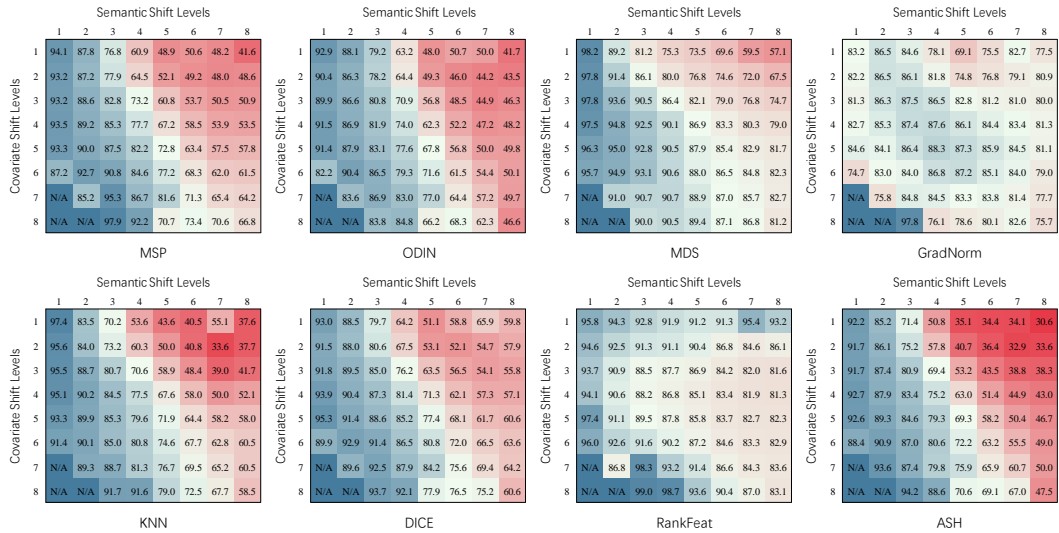

Figure 12: Methods' FPR@95 on all ImageNet-21K subsets with different semantic and covariate shift levels. "N/A" indicates the number of data in this subset is too small for a fair evaluation.

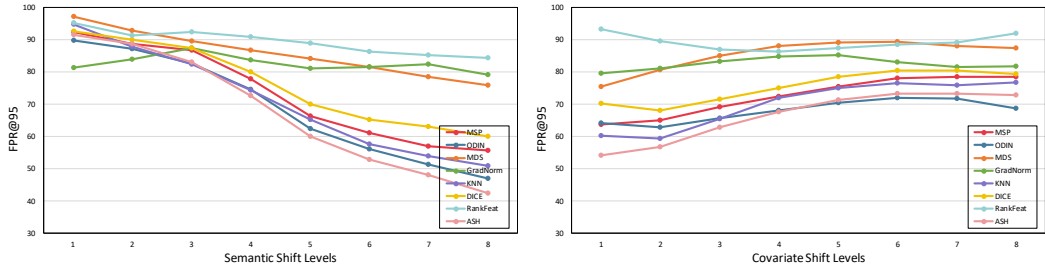

Figure 13: Comparison of methods' FPR@95 across different semantic or covariate shift levels on ImageNet-21K.

Table 4: Results of correlation and sensitivity for FPR@95 on ImageNet-21K

|  | Semantic | | Covariate | |
|---|---|---|---|---|
|  | correlation | sensitivity | correlation | sensitivity |
| MSP [1] | -0.98 | 6.01 | 0.97 | 2.40 |
| ODIN [2] | -0.99 | 6.79 | 0.82 | 1.16 |
| MDS [3] | -1.00 | 2.95 | 0.80 | 1.61 |
| GradNorm [4] | -0.52 | 0.52 | 0.27 | 0.21 |
| KNN [5] | -0.99 | 6.68 | 0.93 | 2.80 |
| DICE [6] | -0.98 | 5.24 | 0.92 | 1.86 |
| RankFeat [7] | -0.97 | 1.52 | -0.06 | 0.06 |
| ASH [8] | -0.99 | 7.75 | 0.94 | 2.95 |

## F.3 FPR@95 Results on Syn-IS

This section presents the FPR@95 results of all the evaluated OOD detection methods on Syn-IS, as shown in Figure 14, Figure 15, and Table 5.

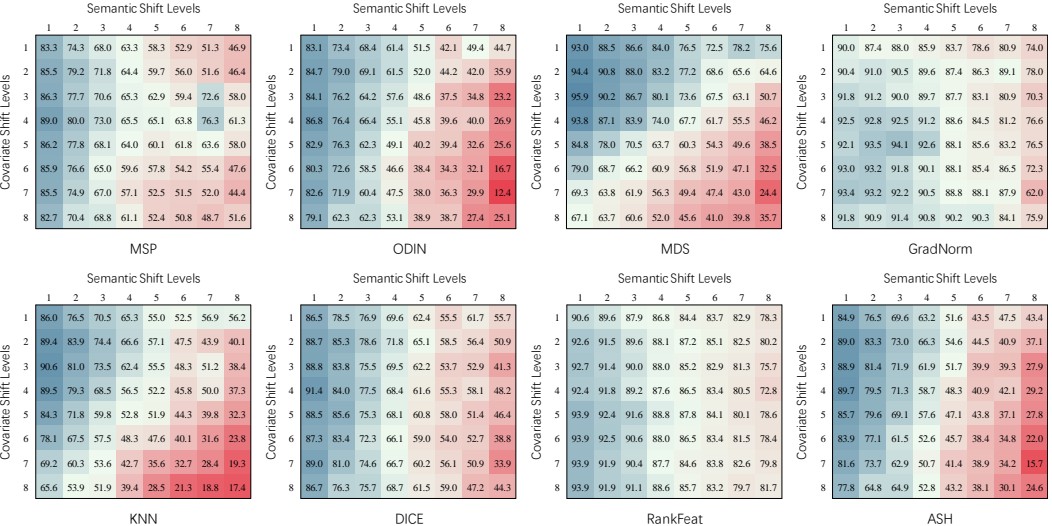

Figure 14: Methods' FPR@95 on all Syn-IS subsets with different semantic and covariate shift levels.

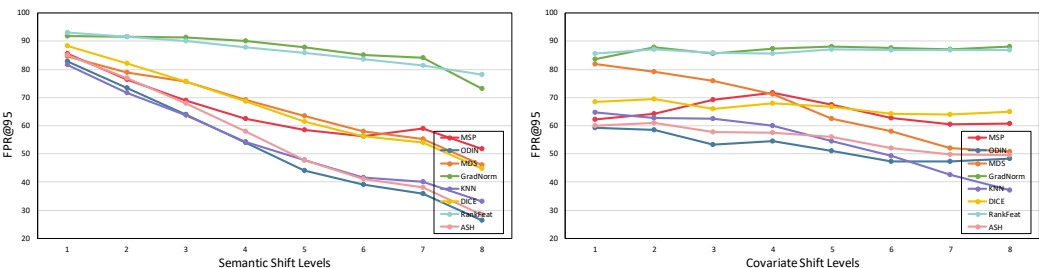

Figure 15: Comparison of methods' FPR@95 across different semantic or covariate shift levels on Syn-IS.

Table 5: Results of correlation and sensitivity for FPR@95 on Syn-IS

|  | **Semantic** | | **Covariate** | |
| --- | --- | --- | --- | --- |
|  | correlation | sensitivity | correlation | sensitivity |
| MSP [1] | -0.94 | 4.36 | -0.37 | 0.62 |
| ODIN [2] | -0.99 | 7.96 | -0.94 | 1.82 |
| MDS [3] | -1.00 | 5.32 | -0.99 | 4.96 |
| GradNorm [4] | -0.88 | 2.24 | 0.64 | 0.42 |
| KNN [5] | -0.99 | 6.79 | -0.97 | 4.05 |
| DICE [6] | -1.00 | 6.09 | -0.83 | 0.68 |
| RankFeat [7] | -0.99 | 2.09 | 0.55 | 0.16 |
| ASH [8] | -0.99 | 8.12 | -0.97 | 1.77 |

## F.4 AUPR Results on ImageNet-21K

This section presents the AUPR results of all the evaluated OOD detection methods on ImageNet-21K, as shown in Figure 16, Figure 17, and Table 6. The results indicate that the AUPR metric is significantly affected by the number of data in the test subsets. Typically, a smaller subset results in a higher AUPR, while a larger subset leads to a lower AUPR. Therefore, the results in this section are not suitable for analyzing the performance of the OOD detection methods.

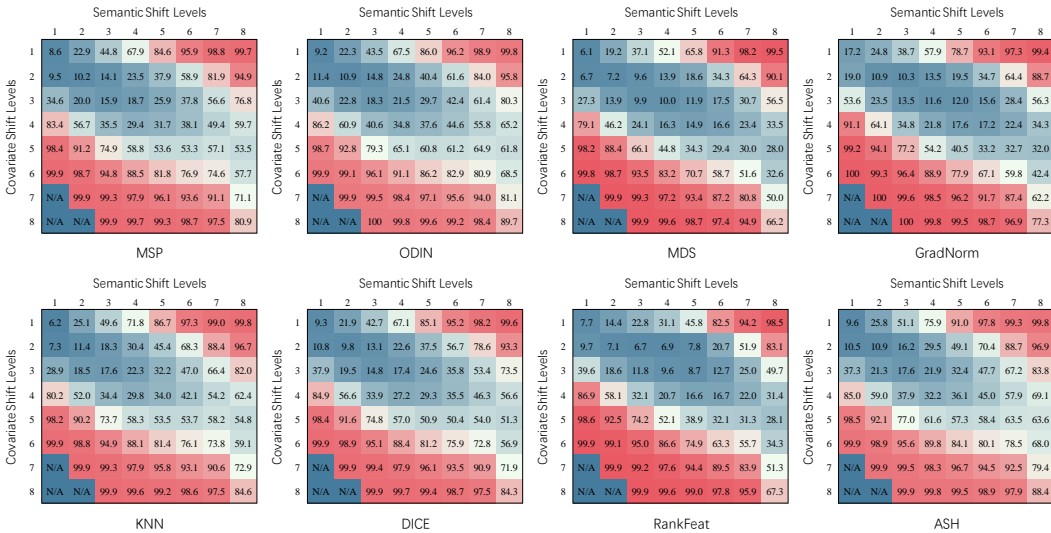

Figure 16: Methods' AUPR on all ImageNet-21K subsets with different semantic and covariate shift levels. "N/A" indicates the number of data in this subset is too small for a fair evaluation.

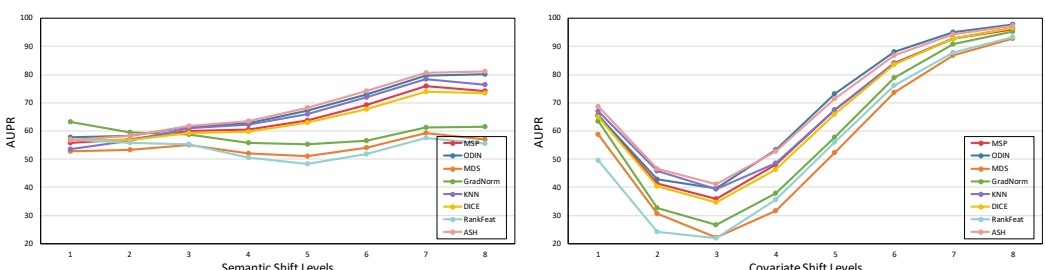

Figure 17: Comparison of methods' AUPR across different semantic or covariate shift levels on ImageNet-21K.

Table 6: Results of correlation and sensitivity for AUPR on ImageNet-21K

|  | **Semantic** | | **Covariate** | |
|---|---|---|---|---|
|  | correlation | sensitivity | correlation | sensitivity |
| MSP [1] | 0.96 | 3.04 | 0.80 | 7.57 |
| ODIN [2] | 0.97 | 3.62 | 0.83 | 7.77 |
| MDS [3] | 0.59 | 0.65 | 0.76 | 8.27 |
| GradNorm [4] | -0.12 | 0.14 | 0.76 | 8.23 |
| KNN [5] | 0.98 | 3.67 | 0.79 | 7.09 |
| DICE [6] | 0.96 | 2.74 | 0.80 | 7.75 |
| RankFeat [7] | -0.13 | 0.17 | 0.84 | 9.61 |
| ASH [8] | 0.98 | 3.85 | 0.80 | 7.09 |

## F.5 AUPR Results on Syn-IS

This section presents the AUPR results of all the evaluated OOD detection methods on Syn-IS, as shown in Figure 18, Figure 19, and Table 7. Since the Syn-IS dataset ensures an equal number of data in each subset during its generation, the bias in AUPR is eliminated. Consequently, the results in this section show a performance trend consistent with the AUROC results described in the main text.

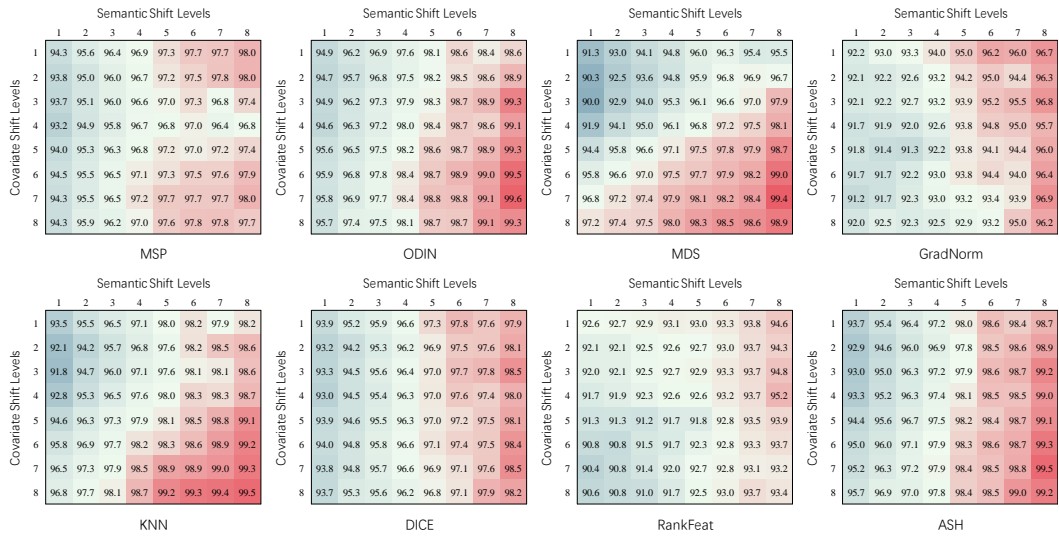

Figure 18: Methods' AUPR on all Syn-IS subsets with different semantic and covariate shift levels.

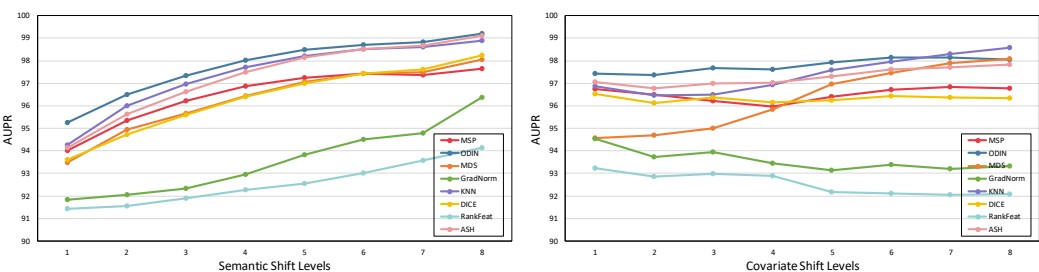

Figure 19: Comparison of methods' AUPR across different semantic or covariate shift levels on Syn-IS.

Table 7: Results of correlation and sensitivity for AUPR on Syn-IS

| | Semantic | | Covariate | |
|---|---|---|---|---|
| | correlation | sensitivity | correlation | sensitivity |
| MSP [1] | 0.91 | 0.47 | 0.38 | 0.05 |
| ODIN [2] | 0.95 | 0.52 | 0.93 | 0.12 |
| MDS [3] | 0.96 | 0.60 | 0.98 | 0.59 |
| GradNorm [4] | 0.97 | 0.63 | -0.82 | 0.16 |
| KNN [5] | 0.93 | 0.60 | 0.93 | 0.31 |
| DICE [6] | 0.98 | 0.63 | 0.08 | 0.00 |
| RankFeat [7] | 0.99 | 0.39 | -0.92 | 0.18 |
| ASH [8] | 0.96 | 0.67 | 0.92 | 0.15 |

# G Further Analysis of Labeling Issues

Leveraging the classification model trained on ImageNet-1K, we analyze the prediction results for the images presented in Figure 2, along with the confidence scores derived using MSP [1], as illustrated in Figure 20. As observed, the model does indeed assign high confidence (i.e., low OOD scores) to the marginal OOD samples, highlighting the limitation of relying solely on semantic labels to distinguish OOD samples.

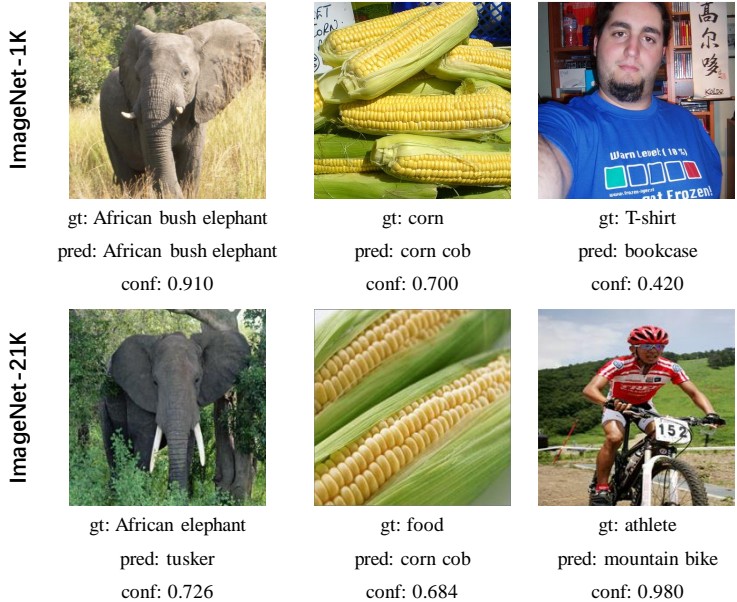

Figure 20: The prediction results and the MSP [1] confidence scores for the marginal OOD images.

We also conduct a quantitative analysis of the labeling issues. We compare the performance of the OOD detection models on the entire ImageNet-21K (with labeling issues) with its performance on subsets with larger semantic shifts (with fewer labeling issues), as shown in Table 8. The results show that models perform better on data with larger semantic shifts. (MSP with AUROC of 79.5 on data with larger semantic shifts and 68.0 on the entire ImageNet-21K.) This result is consistent with the findings in NINCO [14] (manually removes the objects with the labeling issue), where researchers noted that model performance improves when ID-objects are removed from the test set.

Table 8: Results of mean AUROC(%) on subsets with varying degrees of labeling issues

|  | with labeling issues | fewer labeling issues |
|---|---|---|
| MSP [1] | 68.0 | 79.5 |
| ODIN [2] | 74.1 | 84.9 |
| MDS [3] | 52.3 | 58.9 |
| GradNorm [4] | 70.1 | 73.0 |
| KNN [5] | 65.4 | 78.7 |
| DICE [6] | 69.4 | 78.9 |
| RankFeat [7] | 57.3 | 59.0 |
| ASH [8] | 72.3 | 83.7 |

# H Computational Resources

All our experiments are conducted on two A100 GPUs. However, the minimum computational resources required for training the decomposition matrix and evaluating the OOD detection model are much lower than this level; all training and evaluation tasks can be performed on a single RTX 3060 GPU. Although generating the Syn-IS dataset requires more computing power, the actual use

of computational resources is acceptable because we use the SDXL-turbo model to accelerate the generation. Specifically, the generation speed is approximately 400,000 images per day on a single A100 GPU.

# I   Definition of Semantic and Covariate Shift

Here, we give a formal definition of the semantic shift and the covariate shift mentioned in our work. We first define the semantic feature $s$ and the covariate feature $c$ for a sample, assuming that an image $x$ is determined by both $s$ and $c$, while the image label $y$ is influenced only by $s$:

$$
\begin{aligned}
x &= f(s, c), \quad s \in S, c \in C \to x \in X, \\
y &= g(s), \quad s \in S \to y \in Y.
\end{aligned}
\tag{7}
$$

We then provide the definitions for samples with semantic shift and covariate shift.

**Semantic Shift:**
$$
\begin{aligned}
x^{ss} &= f(s', c), \quad s' \notin S, c \in C \to x^{ss} \notin X, \\
y^{ss} &= g(s'), \quad s' \notin S \to y^{ss} \notin Y.
\end{aligned}
\tag{8}
$$

**Covariate Shift:**
$$
\begin{aligned}
x^{cs} &= f(s, c'), \quad s \in S, c' \notin C \to x^{cs} \notin X, \\
y^{cs} &= g(s), \quad s \in S \to y^{cs} \in Y.
\end{aligned}
\tag{9}
$$

In our benchmark, the semantic shift levels measure the distance between the test semantic $s'$ and the ID semantic set $S$. The same applies to the covariate shift levels.

It's worth noting that we observe ImageNet-21K includes labels such as beach and snow, which might serve as backgrounds (covariate contents) in other datasets, but appear as semantic contents in ImageNet-21K.

We treat all the labels in ImageNet-21K (such as backgrounds) as semantic contents in the proposed IS-OOD, as defined in Eq. (7), which might differ from some other benchmarks. In our experiments, the covariate contents only include styles, image augmentations, and lighting changes. However, future researchers can modify the collection of covariate texts according to their needs to study different elements as the covariate components.

# J   Future Directions

In terms of improving the benchmark, a promising direction for future research is to explore the use of more advanced vision-language models, such as BLIP or CoCa, to reduce the gap between text and image feature space, thereby enhancing the accuracy of feature decomposition in the LAID method. Additionally, separate investigations of particular covariate shifts are valuable, such as specifically studying the impact of changes in artistic styles on OOD detection model performance. We also strongly encourage subsequent researchers to use our benchmark for further evaluation of models in other fields, such as assessing whether the understanding of image semantic information by the current large-scale vision-language models, such GPT-4, is affected by image covariate shifts.

# K  Data Sheet

This section provides the dataset documentation for the proposed Syn-IS dataset.

## K.1  Motivation

**For what purpose was the dataset created?** Was there a specific task in mind? Was there a specific gap that needed to be filled? Please provide a description.

- The proposed dataset contains a series of high-quality generated images with more diverse covariate contents to complement the IS-OOD benchmark, which is proposed for evaluating current OOD detection methods.

**Who created the dataset (e.g., which team, research group) and on behalf of which entity (e.g., company, institution, organization)?**

- The Visual Information Processing and Learning (VIPL) research group, Institute of Computing Technology, Chinese Academy of Sciences.

## K.2  Composition

**What do the instances that comprise the dataset represent (e.g., documents, photos, people, countries)?** Are there multiple types of instances (e.g., movies, users, and ratings; people and interactions between them; nodes and edges)? Please provide a description.

- The instances contain objects corresponding to the ImageNet-21K labels.

**How many instances are there in total (of each type, if appropriate)?**

- 5,000 images for each subset, and 64 subsets in total.

**Does the dataset contain all possible instances or is it a sample (not necessarily random) of instances from a larger set?** If the dataset is a sample, then what is the larger set? Is the sample representative of the larger set (e.g., geographic coverage)? If so, please describe how this representativeness was validated/verified. If it is not representative of the larger set, please describe why not (e.g., to cover a more diverse range of instances, because instances were withheld or unavailable).

- No. The dataset are completely generated from scratch.

**What data does each instance consist of?** "Raw" data (e.g., unprocessed text or images) or features? In either case, please provide a description.

- Each instance consists of the image, and the corresponding semantic and covariate shift levels.

**Is there a label or target associated with each instance?** If so, please provide a description.

- No, the Syn-IS dataset is arranged in different folders according to the semantic and covariate shift levels.

**Is any information missing from individual instances?** If so, please provide a description, explaining why this information is missing (e.g., because it was unavailable). This does not include intentionally removed information, but might include, e.g., redacted text.

- No.

**Are relationships between individual instances made explicit (e.g., users' movie ratings, social network links)?** If so, please describe how these relationships are made explicit.

- There are no relationships between individual instances.

**Are there recommended data splits (e.g., training, development/validation, testing)?** If so, please provide a description of these splits, explaining the rationale behind them.

- Yes, the Syn-IS dataset is split according to the semantic and covariate shift levels, which is suited for the evaluation process of our IS-OOD benchmark.

**Are there any errors, sources of noise, or redundancies in the dataset?** If so, please provide a description.

- Errors in image generation resulting are unavoidable. However, we have performed dataset cleaning to minimize these errors.

**Is the dataset self-contained, or does it link to or otherwise rely on external resources (e.g., websites, tweets, other datasets)?** If it links to or relies on external resources, a) are there guarantees that they will exist, and remain constant, over time; b) are there official archival versions of the complete dataset (i.e., including the external resources as they existed at the time the dataset was created); c) are there any restrictions (e.g., licenses, fees) associated with any of the external resources that might apply to a dataset consumer? Please provide descriptions of all external resources and any restrictions associated with them, as well as links or other access points, as appropriate.

- The proposed Syn-IS dose not rely on any external resources.

**Does the dataset contain data that might be considered confidential (e.g., data that is protected by legal privilege or by doctor-patient confidentiality, data that includes the content of individuals' non-public communications)?** If so, please provide a description.

- No.

**Does the dataset contain data that, if viewed directly, might be offensive, insulting, threatening, or might otherwise cause anxiety?** If so, please describe why.

- No.

**Does the dataset relate to people?**

- The Syn-IS dataset may contain images of people, but it is not primarily focused on human subjects.

### K.3 Collection Process

**How was the data associated with each instance acquired?** Was the data directly observable (e.g., raw text, movie ratings), reported by subjects (e.g., survey responses), or indirectly inferred/derived from other data (e.g., part-of-speech tags, model based guesses for age or language)? If data was reported by subjects or indirectly inferred/derived from other data, was the data validated/verified? If so, please describe how.

- Data are generated with prompts and associated with the label in the prompts.

**What mechanisms or procedures were used to collect the data (e.g., hardware apparatus or sensor, manual human curation, software program, software API)?** How were these mechanisms or procedures validated?

- We collect the data by software program.

**If the dataset is a sample from a larger set, what was the sampling strategy (e.g., deterministic, probabilistic with specific sampling probabilities)?**

- No.

**Who was involved in the data collection process (e.g., students, crowdworkers, contractors) and how were they compensated (e.g., how much were crowdworkers paid)?**

- The images are generated by software program (SDXL).

**Over what timeframe was the data collected? Does this timeframe match the creation timeframe of the data associated with the instances (e.g., recent crawl of old news articles)?** If not, please describe the timeframe in which the data associated with the instances was created.

- Our dataset is constructed in April of 2024.

**Were any ethical review processes conducted (e.g., by an institutional review board)?** If so, please provide a description of these review processes, including the outcomes, as well as a link or other access point to any supporting documentation.

- No.

**Did you collect the data from the individuals in question directly, or obtain it via third parties or other sources (e.g., websites)?**

- No.

**Were the individuals in question notified about the data collection?** If so, please describe (or show with screenshots or other information) how notice was provided, and provide a link or other access point to, or otherwise reproduce, the exact language of the notification itself.

- N/A. Our dataset does not involve the collection from the individuals.

**Did the individuals in question consent to the collection and use of their data?** If so, please describe (or show with screenshots or other information) how consent was requested and provided, and provide a link or other access point to, or otherwise reproduce, the exact language to which the individuals consented.

- N/A. Our dataset does not involve the collection from the individuals.

**If consent was obtained, were the consenting individuals provided with a mechanism to revoke their consent in the future or for certain uses?** If so, please provide a description, as well as a link or other access point to the mechanism (if appropriate).

- N/A. Our dataset does not involve the collection from the individuals.

**Has an analysis of the potential impact of the dataset and its use on data subjects (e.g., a data protection impact analysis) been conducted?** If so, please provide a description of this analysis, including the outcomes, as well as a link or other access point to any supporting documentation.

- No.

### K.4 Preprocessing/cleaning/labeling

**Was any preprocessing/cleaning/labeling of the data done (e.g., discretization or bucketing, tokenization, part-of-speech tagging, SIFT feature extraction, removal of instances, processing of missing values)?** If so, please provide a description. If not, you may skip the remaining questions in this section.

- No.

**Was the "raw" data saved in addition to the preprocessed/cleaned/labeled data (e.g., to support unanticipated future uses)?** If so, please provide a link or other access point to the "raw" data.

- Yes. It can be downloaded at our code repo.

**Is the software that was used to preprocess/clean/label the data available?** If so, please provide a link or other access point.

- N/A.

### K.5 Uses

**Has the dataset been used for any tasks already?** If so, please provide a description.

- No.

**Is there a repository that links to any or all papers or systems that use the dataset?** If so, please provide a link or other access point.

- N/A.

**What (other) tasks could the dataset be used for?**

- This dataset could potentially be used for classification tasks.

**Is there anything about the composition of the dataset or the way it was collected and preprocessed/cleaned/labeled that might impact future uses?** For example, is there anything that a dataset consumer might need to know to avoid uses that could result in unfair treatment of individuals or groups (e.g., stereotyping, quality of service issues) or other risks or harms (e.g., legal risks, financial harms)? If so, please provide a description. Is there anything a dataset consumer could do to mitigate these risks or harms?

- No.

**Are there tasks for which the dataset should not be used?** If so, please provide a description.

- No.

### K.6 Distribution

**Will the dataset be distributed to third parties outside of the entity (e.g., company, institution, organization) on behalf of which the dataset was created?** If so, please provide a description.

- Yes. Any

**How will the dataset will be distributed (e.g., tarball on website, API, GitHub)?** Does the dataset have a digital object identifier (DOI)?

- We will open-source our dataset on our GitHub project homepage. At the moment, we do not have a DOI number.

**When will the dataset be distributed?**

- The dataset can be downloaded right now.

**Will the dataset be distributed under a copyright or other intellectual property (IP) license, and/or under applicable terms of use (ToU)?** If so, please describe this license and/or ToU, and provide a link or other access point to, or otherwise reproduce, any relevant licensing terms or ToU, as well as any fees associated with these restrictions.

- The licence of Syn-IS is "CreativeML Open RAIL++-M", which follows the licence set by the Stable Diffusion XL.

**Have any third parties imposed IP-based or other restrictions on the data associated with the instances?** If so, please describe these restrictions, and provide a link or other access point to, or otherwise reproduce, any relevant licensing terms, as well as any fees associated with these restrictions.

- No.

**Do any export controls or other regulatory restrictions apply to the dataset or to individual instances?** If so, please describe these restrictions, and provide a link or other access point to, or otherwise reproduce, any supporting documentation.

- Not yet.

### K.7 Maintenance

**Who will be supporting/hosting/maintaining the dataset?**

- The Visual Information Processing and Learning (VIPL) research group.

**How can the owner/curator/manager of the dataset be contacted (e.g., email address)?**

- zhangjie@ict.ac.cn

**Is there an erratum?** If so, please provide a link or other access point.

- No.

**Will the dataset be updated (e.g., to correct labeling errors, add new instances, delete instances)?** If so, please describe how often, by whom, and how updates will be communicated to dataset consumers (e.g., mailing list, GitHub)?

- There are no plans at the moment, but if there are updates, they will be announced, and the download source will be updated on the project homepage.

**If the dataset relates to people, are there applicable limits on the retention of the data associated with the instances (e.g., were the individuals in question told that their data would be retained for a fixed period of time and then deleted)?** If so, please describe these limits and explain how they will be enforced.

- No.

**Will older versions of the dataset continue to be supported/hosted/maintained?** If so, please describe how. If not, please describe how its obsolescence will be communicated to dataset consumers.

- Yes. If there are any updates, the previous version of the dataset will also be shared on website for download.

**If others want to extend/augment/build on/contribute to the dataset, is there a mechanism for them to do so?** If so, please provide a description. Will these contributions be validated/verified? If so, please describe how. If not, why not? Is there a process for communicating/distributing these contributions to dataset consumers? If so, please provide a description.

- Yes. We welcome and encourage researchers to extend/augment/build on/contribute to our dataset for non-profit purposes without the need for prior notification.

