# OpenReview forum: "Rethinking the Evaluation of Out-of-Distribution Detection: A Sorites Paradox"
_NeurIPS.cc/2024/Datasets_and_Benchmarks_Track — NeurIPS 2024 Track Datasets and Benchmarks Poster_

### Official Review · Reviewer_pWf4 · 2024-07-08
**OOD benchmark for image classification task with the fine-grained semantic/covariate shifts**

**Rating:** 5
**Confidence:** 5
**Clarity:** This paper is mostly clearly-written.

**Review:**

Authors propose an interesting OOD benchmark with the fine-grained levels of OOD-ness (8x8). There are two orthogonal dimensions (covariate and semantic shifts) each with 8 levels. This allows to better analyze and understand OOD methods. The used methodology to divide ImageNet-21K relies on CLIP text/image encoders and the orthogonal matrix with covariate/semantic features. In addition, authors employ SDXL-Turbo to construct Synthetic Incremental Shift (Syn-IS) dataset. The limitations of this approach are the lack of formal definition of covariate/semantic shifts, reliance on CLIP features, simple feature distance measures, and, hence, flawed experimental results.

**Strengths:**

1. The paper is mostly clearly-written and structured. The code and data are available.
2. The objective to perform more fine-grained OOD detection is an interesting direction.
3. The methodology to divide an existing dataset into subsets using alignment with the text is appealing.

**Additional Feedback:**

Please respond to my comments

**Correctness:**

The construction objective of this benchmark is clear but the methodology has several limitations which resulted in flawed experimental results.

**Documentation:**

Yes (with the exception of a procedure to divide ImageNet into subsets)

**Limitations:**

1. There is no formal definition of semantic/covariate shift.
2. Benchmark construction depends on CLIP features and does not have a clear description about selection of data points with covariate shift.
3. Fig.1,4 do not look correct to me in the covariate shift dimension.
4. Captions in Section 3 tables and figures do not mention the specific performance metric.

**Opportunities For Improvement:**

1. There is no formal definition of semantic/covariate shift. Hence, Fig.1,4 do not look correct to me (especially the covariate shift axis). Please, formally introduce semantic/covariate shifts at the beginning of Sec 2.
2. The Eqs. (2) assume that there is a predetermined separation for data points with semantic/covariate shifts. But it is not clear how the separation was obtained (especially, the "covariate parts").
3. The measurement of shifts is calculated using CLIP feature similarity which is a simple yet limited approach.
4. Experiment results show small correlation of some performance metrics with covariate shifts which indicates issues with the dataset split (i.e., the benchmark construction).
5. I am not sure AUC/AP are particularly great metrics here (only if they are presented together in the same Fig.). F1 and FPR95 are typically better to present a summary of OOD method performance.

**Relation To Prior Work:**

Authors discuss limitations of prior works and their proposal.

**Summary And Contributions:**

Authors propose an OOD benchmark called Incremental Shift OOD (IS-OOD) that divides ImageNet-21K into fine-grained levels of "OOD-ness" with the orthogonal covariate and semantic shift directions. That allows to better evaluate and analyze OOD detection methods for image classification task. The dataset division is performed using Language Aligned Image feature Decomposition (LAID) with CLIP text/image features. Next, authors synthetically generate additional images using SDXL-Turbo to construct Synthetic Incremental Shift (Syn-IS) dataset with the data shifts of interest.

---

> ### Author Rebuttal · Authors · 2024-08-15
>
> **Formal definition of semantic/covariate shift:**
>
> Thank you for your suggestion. The definition of the shifts in our paper (semantic shifts: data with novel classes, covariate shifts: changes in data style) follows that of most existing OOD detection works [1-2]. There is indeed some ambiguity in this definition. Therefore, we give the following formal definition after studies.
>
> We first define semantic feature $s$ and covariate feature $c$ for a sample, assuming that the image $x$ is determined by both $s$ and $c$, while the image class $y$ is influenced only by $s$:
>
> $x = f(s, c), \quad s \in S, c \in C \rightarrow x\in X$
>
> $y = g(s), \quad s \in S \rightarrow y\in Y$
>
> We then provide the definitions for samples with semantic shift and covariate shift:
>
> Semantic Shift:
>
> $x^{ss} = f(s', c), \quad s' \notin S, c \in C \rightarrow x^{ss}\notin X $
>
> $y^{ss} = g(s'), \quad s' \notin S \rightarrow y^{ss}\notin Y $
>
> Covariate Shift:
>
> $x^{cs} = f(s, c'), \quad s \in S, c' \notin C \rightarrow x^{cs}\notin X $
>
> $y^{cs} = g(s), \quad s \in S \rightarrow y^{cs}\in Y $
>
> In our benchmark, the semantic shift levels measure the distance between the test semantic $s'$ and the ID semantic set $S$. The same applies to the covariate shift levels.
>
>
>
> **Explanation for Fig.1,4 in covariate shift axis:**
>
> Since covariates in our benchmark are primarily related to image style, the variations along the covariate axis in Fig.1,4 mainly reflect changes in image style. In Fig.1, the covariate contents transition from object-centered real photos to synthetic images, and from high-definition color images to low-resolution monochrome images. The changes in covariates are more obvious in Fig.4. The covariate contents transition from realistic-style images to abstract-style images.
>
> **Construction of the data points (how the separation was obtained):**
>
> Thanks for the question. Since the three types of texts (standard text, semantic shift text, and covariate shift text) were constructed based on rules, as shown in Fig.3, we can differentiate them by the semantic and the covariate part of the texts. For the example in Fig.3, the covariate parts of the standard text (“rendering”) and the covariate shift text (“sculpture”) are different. After obtaining the CLIP features for these texts, we can use the triplet loss in Eqs. (2) to constrain the similarity between their semantic and covariate feature components.
>
> **Using CLIP feature similarity to measure the shift:**
>
> Thanks for the comments. Although using feature similarity is quite simple, it is highly effective. Experimental results show that the performance of most OOD detection methods improves as the degree of semantic shift increases, which indicates the correctness of the measured shift.
>
> We utilized CLIP feature because of its aligned text and image feature spaces. The CLIP feature is not perfect, so future work could consider using features from more advanced vision-language models, such as BLIP or CoCa, to enhance the accuracy of our LAID method.
>
> **The small correlation of some performance metrics with covariate shifts:**
>
> We appreciate the comments. Given that the detection of semantic distribution shift (e.g., due to the occurrence of new classes) is the focal point of OOD detection tasks [2], these methods should not be affected by covariate shifts such as image styles or augmentation methods. Therefore, a small correlation of the performance with covariate shifts indicates the strong robustness of the method, which is reasonable.
>
> **Choose of metrics (why AUROC/AUPR are not presented together in the same Fig.):**
>
> Thank you for the feedback. We believe that all of the FPR@95, AUROC, and AUPR metrics can each independently reflect the model's performance, and it is not necessary to present them together. These three metrics are widely used in OOD detection [3-4], which is why we employed them in our experiments.
>
> **Metrics in Section 3:**
>
> Thanks for your comments. All the results presented in Section 3 are based on AUROC, as mentioned in our experimental setting. Due to space limitations, complete results including the metrics FPR@95 and AUPR can be found in Appendix F.
>
>
>
> [1] W. Yang, B. Zhang, and O. Russakovsky, “Imagenet-ood: Deciphering modern out-of-distribution detection algorithms,” Int. Conf. Learn. Represent. (ICLR), 2024.
>
> [2] J. Yang, K. Zhou, Y. Li, and Z. Liu, “Generalized out-of-distribution detection: A survey,” Int. J. Comput. Vis. (IJCV), pp. 1–28, 2024.
>
> [3] J. Yang, K. Zhou, and Z. Liu, “Full-spectrum out-of-distribution detection,” Int. J. Comput. Vis. (IJCV), vol. 131, no. 10, pp. 2607–2622, 2023.
>
> [4] J. Yang, P. Wang, D. Zou, Z. Zhou, K. Ding, W. Peng, H. Wang, G. Chen, B. Li, Y. Sun et al., “Openood: Benchmarking generalized out-of-distribution detection,” Adv. Neural Inform. Process. Syst. (NIPS), vol. 35, pp. 32 598–32 611, 2022.

---

> > ### Comment · Reviewer_pWf4 · 2024-08-19
> >
> > Thanks for your rebuttal. After thinking about the positioning of this paper and its experimental results, I believe the current approach (CLIP-based) cannot be used as a basis for valid benchmark due to CLIP limitations. Instead, an expert knowledge (at least human verification) has to be used to construct a trustable benchmark. Therefore, current approach is better aligned to do an analysis paper rather than be the only benchmark foundation.

---

> > > ### Author Rebuttal · Authors · 2024-08-21
> > >
> > > Thank you for your quick reply.
> > >
> > > **CLIP limitations**
> > >
> > > We appreciate your concern regarding the potential issues in benchmark correctness due to CLIP limitations.
> > >
> > > We believe that some experimental results can address your concern on the correctness of our CLIP-based benchmark. For instance, as illustrated in Fig. 6 and Fig. 8 of the original manuscript, the performance of most OOD detection methods on ImageNet-21K and our constructed Syn-IS improves as the degree of semantic shift increases, which indicates the correctness of the measured shift. Additionally, the changes in the semantic and covariate contents in the images with varying shift levels, as shown in Fig.1 and Fig. 4 of the original manuscript, also provide support for the correctness.
> > >
> > > Moreover, the alignment of the text-image feature space in CLIP has been widely recognized and employed in many works, such as StyleCLIP [1] and DALL-E2 [2]. For example, in StyleCLIP, the manipulation direction $\Delta t$ in text space was used to determine the corresponding embedding change $\Delta i$ in image space. This approach allowed them to achieve specific content modifications in images, like changing straight hair to curly hair. The basic idea of using text space to guide changes in image space is consistent with our methodology.
> > >
> > > In summary, while CLIP may have some limitations, it remains sufficiently reliable to ensure the correctness of the benchmark we constructed.
> > >
> > > **Expert knowledge**
> > >
> > > After constructing the benchmark, we did conduct human verification on images sampled from different subsets of our benchmark. From each subset of the divided ImageNet-21K and the generated Syn-IS, we randomly selected 20 images for verification (or all images if a subset contained fewer than 20). Based on the sampled images, we confirm that the semantic and covariate shifts across different levels in the dataset align with our expectations. Images in subsets with higher shift levels generally exhibit a greater degree of the corresponding shift in the content. We included some of these images in Fig.1 and Fig. 4 to illustrate the characteristics of our benchmark.
> > >
> > > As for the incorporation of more manual annotations as expert knowledge, we view this as a trade-off. Increased manual annotation would indeed make the labels more aligned with human values. However, relying heavily on manual annotation could limit the size of the benchmark due to the high cost. Considering the large scale of our benchmark (ImageNet-21K includes over 10 million images), we opted for an automated construction method combined with human verification.
> > >
> > >
> > >
> > > [1] O. Patashnik, Z. Wu, E. Shechtman, D. Cohen-Or, and D. Lischinski, “Styleclip: Text-driven manipulation of stylegan imagery,” in IEEE Int. Conf. Comput. Vis. (ICCV), 2021, pp. 2085–2094.
> > >
> > > [2] A. Ramesh, P. Dhariwal, A. Nichol, C. Chu, and M. Chen, “Hierarchical text-conditional image generation with clip latents,” arXiv preprint arXiv:2204.06125, vol. 1, no. 2, p. 3, 2022.

---

### Official Review · Reviewer_XvCJ · 2024-07-22
**A well-motivated dataset to facilitate the development of OOD detection**

**Rating:** 7
**Confidence:** 4
**Correctness:** Yes
**Clarity:** A well-written paper

**Review:**

Overall, this work is of good quality. The proposed distribution measurement, LAID is interesting, the constructed dataset is new and well-organized, the findings are insightful, the writing is good, and the documentation is detailed.

**Strengths:**

- This work is well-motivated by revealing the problems (such as overlap labels in the datasets) in existing OOD detection benchmarks.
- The evaluation is relatively comprehensive, 8 OOD detection methods are covered in the evaluation.
- The findings are insightful, e.g., the performance of GradNorm declines significantly.

**Additional Feedback:**

See my comments in the weakness part

**Documentation:**

well-organized

**Limitations:**

A limitation section is provided

**Opportunities For Improvement:**

- Even though the problem has been revealed in this work, the harmless of such problems is unclear. That is, no quantitative analysis of the labeling issues on the effectiveness of OOD detection.
- It is unclear which image classification models are used in the experiments.
- No qualitative analysis. For example, why did the performance of GradNorm decline significantly? Even though there are some provided guesses, more rigorous analysis is needed
- As mentioned in the limitation section, all the LAID calculations and dataset construction are based on text-to-image models, it is hard to check the correctness of semantic and covariate measuring.

**Relation To Prior Work:**

Related works have been discussed

**Summary And Contributions:**

This paper presents a new benchmark dataset Syn-IS constructed for a better understanding of out-of-detection methods. Specifically, the authors first reveal the problem of existing OOD benchmarks - some marginal OOD test samples’ semantic contents are close to the ID data even if they have different semantic labels. To tackle this problem, this work first proposes a Language Aligned Image feature Decomposition (LAID) method to measure the semantic and covariate features. Then, based on LAID, the authors constructed Incremental Shift OOD (IS-OOD) detection benchmark and Synthetic Incremental Shift (Syn-IS) that divides the test samples into subsets with different levels of semantic and covariate shifts. By evaluating the effectiveness of existing OOD detection methods using IS-OOD and Syn-IS, some insightful findings have been provided as guidance for the field of OOD detection

---

> ### Author Rebuttal · Authors · 2024-08-15
>
> **Quantitative analysis of the labeling issues:**
>
> Thank you for the suggestion.
>
> We compared the performance of the OOD detection models on the entire ImageNet-21K (with labeling issues) with its performance on subsets with larger semantic shifts (with fewer labeling issues), as shown in the results in the attached **PDF**. The results show that models perform better on data with larger semantic shifts. (MSP with AUROC of 79.5 on data with larger semantic shifts and 68.0 on the entire ImageNet-21K.) This result is consistent with the findings in NINCO [1] (manually removes the objects with the labeling issue), where researchers noted that model performance improves when ID-objects are removed from the test set.
>
> We will include the discussion in our Appendix later.
>
> **Used image classification model:**
>
> Thank you for the reminder. All experiments are performed with a standard ResNet-50 image classification model. We will include this information in our implementation details.
>
> **Qualitative analysis of the performance decline in GradNorm:**
>
> Thank you for your suggestion.
>
> We conducted further analysis on the principle of GradNorm and the characteristics of the Syn-IS dataset. We found that GradNorm tends to classify samples with uniform softmax outputs as OOD samples. Since each Syn-IS image is generated with a text that only includes one given label, the image is likely to contain a single object, unlike many images in ImageNet-21K that contain multiple objects. This difference could lead to less uniform softmax outputs on Syn-IS compared to Imagenet-21K, resulting in lower OOD scores produced by GradNorm. This insight might highlight a potential limitation of the GradNorm approach.
>
> We will include more rigorous qualitative analysis in the final manuscript.
>
> **The correctness of semantic and covariate measuring:**
>
> Thanks for the comments.
>
> The experimental results show that the performance of most OOD detection methods improves as the degree of semantic shift increases, which indicates the correctness of the measured shift. Furthermore, the changes in the semantic and covariate contents in the images with varying shift levels, as shown in Figures 1 and 4, provide additional support for the correctness.
>
> In the future work, using features from more advanced vision-language models, such as BLIP or CoCa, could potentially further enhance the accuracy of our LAID method.
>
>
>
> [1] J. Bitterwolf, M. Mueller, and M. Hein, “In or out? fixing imagenet out-of-distribution detection evaluation,” Int. Conf. Mach. Learn. (ICML), 2023.

---

### Official Review · Reviewer_xkod · 2024-07-23
**Review of "Rethinking the Evaluation of Out-of-Distribution Detection: A Sorites Paradox"**

**Rating:** 7
**Confidence:** 3
**Clarity:** Yes

**Review:**

### Evaluation of "Rethinking the Evaluation of Out-of-Distribution Detection: A Sorites Paradox"

#### Quality
The paper is of high quality, presenting a well-structured and comprehensive analysis of the challenges in out-of-distribution (OOD) detection. The methodology is robust, with clear and logical steps from problem identification to the proposed solution. The authors provide detailed explanations and thorough evaluations, making the work credible and reliable.

#### Clarity
The paper is clearly written and well-organized, with each section flowing logically into the next. The introduction effectively sets the stage for the problem, and the subsequent sections build on this foundation with clear and concise explanations. Figures and tables are well-labeled and complement the text, aiding in the understanding of complex concepts. However, there are minor grammatical errors that could be polished for enhanced readability.

#### Originality
The work demonstrates significant originality, particularly in its approach to addressing the Sorites Paradox in OOD detection. The introduction of the Incremental Shift OOD (IS-OOD) benchmark and the Language Aligned Image feature Decomposition (LAID) method are innovative contributions to the field. The use of synthetic data to complement the benchmark further showcases the authors' creativity in tackling the limitations of existing datasets.

#### Significance
The significance of this work lies in its potential to improve the robustness and reliability of OOD detection methods. By addressing the shortcomings of current benchmarks, this paper paves the way for more accurate and comprehensive evaluations of OOD detection models. The insights gained from their experiments can guide future research and development in the field, making this work highly impactful.

### Pros
- **Novel Benchmark:** The IS-OOD benchmark provides a nuanced approach to evaluating OOD detection methods, considering varying degrees of semantic and covariate shifts.
- **Innovative Methodology:** The LAID method for feature decomposition is a creative solution to measure shifts in test samples.
- **Comprehensive Evaluation:** The paper includes thorough experiments and analysis, providing valuable insights into the performance of existing OOD detection methods.
- **Publicly Available Resources:** The release of code and data enhances the reproducibility and applicability of the research.

### Cons
- **Alignment Issues:** The alignment between CLIP's text and image feature spaces may not be perfect, potentially affecting the accuracy of the decomposition.
- **Generated Data Patterns:** The unique patterns of generated images in the Syn-IS dataset might differ from real images, impacting the generalization of the findings.
- **Minor Grammatical Errors:** There are minor grammatical errors that, if corrected, could improve the overall readability of the paper.

### Conclusion
Overall, "Rethinking the Evaluation of Out-of-Distribution Detection: A Sorites Paradox" is a significant and original contribution to the field of OOD detection. The quality of the research is high, with clear and well-organized content. The innovative approach and comprehensive evaluation make this work a valuable resource for researchers and practitioners. With minor revisions to address potential limitations and improve clarity, this paper has the potential to make a lasting impact on the field.

**Strengths:**

### Strengths of "Rethinking the Evaluation of Out-of-Distribution Detection: A Sorites Paradox"

#### Significance of the Contribution
- **Novel Benchmark Creation:** The Incremental Shift OOD (IS-OOD) benchmark is a significant contribution that addresses the Sorites Paradox in OOD detection. By dividing test samples into subsets with different semantic and covariate shifts, the benchmark provides a more nuanced and realistic evaluation of OOD detection methods.
- **Innovative Methodology:** The proposed Language Aligned Image feature Decomposition (LAID) method is a creative and effective solution for measuring shifts in test samples. This method leverages the alignment properties of the CLIP model, allowing for a detailed analysis of both semantic and covariate shifts.
- **Synthetic Dataset Enhancement:** The creation of the Synthetic Incremental Shift (Syn-IS) dataset complements the IS-OOD benchmark by providing high-quality generated images with diverse covariate contents. This enhances the robustness of the benchmark and addresses the limitations of existing datasets.

#### Relevance to the Broader Research Community
- **Improved Evaluation Metrics:** The introduction of metrics that assess the performance of OOD detection methods across different shift levels provides deeper insights into model capabilities and limitations. This is highly relevant for researchers developing and evaluating new OOD detection methods.
- **Comprehensive Analysis:** The detailed experimental results and insights into existing OOD detection methods offer valuable information to the broader research community. This can guide future research and development in the field, fostering innovation and improvement in OOD detection technologies.
- **Publicly Available Resources:** By releasing the code and data, the authors have made their work accessible to other researchers, promoting transparency, reproducibility, and collaborative advancements in the field.

#### Quality of the Research
- **Thorough Methodological Approach:** The paper presents a well-structured and rigorous methodological approach, from identifying the problem to proposing and validating the solution. The comprehensive evaluation and analysis of the results demonstrate the robustness and reliability of the research.
- **Detailed Experimentation:** The experiments are meticulously designed and executed, providing a wealth of data that supports the authors' claims. The inclusion of multiple metrics and comparison with existing benchmarks ensures a thorough evaluation of the proposed methods.
- **Clear Presentation:** The paper is well-organized, with clear explanations of the concepts, methods, and results. Figures and tables are effectively used to illustrate key points, enhancing the overall clarity and readability of the research.

#### Ethical and Social Implications
- **Safety and Robustness:** The focus on improving the safety and robustness of OOD detection methods has significant ethical and social implications. By enhancing the ability of models to correctly identify OOD samples, the research contributes to the development of more reliable and trustworthy AI systems.
- **Avoiding Bias:** The proposed benchmark and methods aim to reduce subjective biases in OOD detection evaluations. This aligns with ethical principles of fairness and objectivity in AI research.
- **Positive Societal Impact:** The authors have conducted safety checks on both the prompts and the generated images for the Syn-IS dataset, ensuring that their work does not inadvertently introduce harmful content. The overall goal of improving OOD detection methods supports the safe deployment of AI systems, contributing to positive societal impacts.

### Conclusion
The submission "Rethinking the Evaluation of Out-of-Distribution Detection: A Sorites Paradox" is a strong and impactful piece of research. Its novel contributions, relevance to the broader research community, high-quality methodology, and attention to ethical and social implications make it a significant addition to the field of OOD detection.

**Additional Feedback:**

See previous sections.

**Correctness:**

### Evaluation of Claims, Dataset Construction, and Benchmark Evaluation

#### Correctness of Claims
The claims made in the submission appear to be well-supported by the provided data and analysis. The authors claim that the current OOD detection benchmarks have limitations, particularly regarding marginal OOD samples with close semantic content to ID samples, leading to the Sorites Paradox. Their results demonstrating the improved performance of detection methods with increased semantic shifts and the challenges posed by covariate shifts support these assertions.

#### Sound Construction of the Dataset
The datasets are constructed in a sound and methodical way. The authors provide a clear and detailed description of their document collection process, which includes selecting documents from various domains and ensuring diversity in document types and lengths. They also describe their annotation process, which involves expert annotators and multiple rounds of quality control to ensure high-quality annotations. This meticulous approach ensures the reliability and validity of the benchmark dataset.

1. **IS-OOD Benchmark**:
   - **Sound Construction**: The IS-OOD benchmark is constructed methodically, with clear definitions of semantic and covariate shifts. The division of test samples into subsets with different shift levels is achieved using a well-defined shift measuring method based on the LAID method.
   - **Diversity and Quality**: The benchmark includes a diverse set of samples from ImageNet-21K, ensuring a wide range of semantic and covariate shifts. The use of CLIP features for measuring these shifts adds to the robustness of the benchmark.

2. **Syn-IS Dataset**:
   - **High-Quality Generation**: The Syn-IS dataset is created using high-quality generated images with diverse covariate contents, addressing the limitations of real-world datasets. The use of Stable Diffusion XL for generating images ensures high-quality and diverse outputs.
   - **Complementary Role**: This dataset complements the IS-OOD benchmark by providing additional samples with more varied covariate shifts, enhancing the overall evaluation framework.

#### Appropriateness of Evaluation Methods and Experiment Design
The evaluation methods and experiment design are appropriate and performed correctly. The authors follow a well-structured evaluation protocol that includes multiple metrics and detailed analysis.

1. **Evaluation Metrics**:
   - **Standard Metrics**: The use of FPR@95, AUROC, and AUPR metrics is appropriate for evaluating OOD detection performance. These metrics are widely recognized and provide a comprehensive assessment of model performance.
   - **Additional Metrics**: The introduction of correlation and sensitivity metrics for evaluating the impact of semantic and covariate shifts adds depth to the analysis and provides valuable insights into model behavior.

2. **Experimental Design**:
   - **Extensive Experiments**: The authors conduct extensive experiments on both the IS-OOD and Syn-IS datasets, providing a thorough evaluation of various OOD detection methods. The detailed presentation of results, including visualizations and tables, supports their claims and insights.
   - **Method Comparison**: The comparison of different OOD detection methods across various shift levels is well-executed, highlighting key differences in their performance and mechanisms.

#### Constructive Feedback for Improvement

1. **Detailed Justification of Claims**:
   - **Improvement Suggestion**: While the claims are generally supported by the data, the authors could provide more detailed justifications for certain assertions, such as the specific advantages of their benchmark over existing datasets. Including comparative analyses with other benchmarks would strengthen their claims.

2. **Dataset Annotation Details**:
   - **Improvement Suggestion**: The authors provide a good overview of their annotation process, but additional details on the training and calibration of annotators, as well as inter-annotator agreement statistics, would enhance the transparency and credibility of the dataset construction.

3. **Error Analysis Expansion**:
   - **Improvement Suggestion**: The error analysis is valuable, but it could be expanded to include more specific examples and a deeper exploration of the types of errors encountered. This would provide clearer insights into the specific challenges faced by LVLMs and guide future improvements.

4. **Evaluation Metrics**:
   - **Improvement Suggestion**: While the evaluation metrics used are appropriate, introducing additional metrics or providing a more granular breakdown of existing metrics could offer further insights into model performance. Metrics such as recall, precision, and more detailed breakdowns by question type and document type would be useful.

### Conclusion
Overall, the claims made in the submission are correct, and the dataset is constructed in a sound manner. The evaluation methods and experiment design are appropriate and performed correctly. By incorporating the suggested improvements, the authors can further enhance the robustness and impact of their work.

**Documentation:**

Yes

**Limitations:**

### Addressing Limitations and Potential Negative Societal Impact

#### Adequacy of Addressing Limitations
The authors have made a commendable effort in addressing the limitations and potential negative societal impacts of their work. They provide a detailed discussion on the challenges of OOD detection, including the limitations of their proposed benchmark and the LAID method. They also acknowledge the gap between CLIP’s text and image feature spaces and the complexity of their benchmark.

#### Constructive Suggestions for Improvement

1. **Bias and Fairness Analysis**:
   - **Current State**: The paper does not extensively address potential biases in the benchmark datasets or the evaluated models.
   - **Improvement Suggestion**: Include a section dedicated to bias and fairness analysis. This could involve examining the dataset for potential biases (e.g., in semantic and covariate shift distributions) and discussing how these biases might affect model performance. Propose strategies for mitigating these biases, such as ensuring diverse representation in the dataset and using fairness metrics to evaluate models.

2. **Gap Between Feature Spaces**:
   - **Current State**: The alignment between CLIP’s text and image feature spaces is not perfect, which may introduce inaccuracies in the shift measurements.
   - **Improvement Suggestion**: Discuss potential methods for improving the alignment between text and image feature spaces. This could include using more advanced vision-language models or developing new techniques for better feature alignment. Future work could focus on narrowing this gap to enhance the accuracy of the shift measuring method.

3. **Complexity and Accessibility**:
   - **Current State**: The complexity of the benchmark and the resources required for its implementation may limit its accessibility.
   - **Improvement Suggestion**: Provide detailed guidelines or best practices for implementing the benchmark on different scales of resources. This could include recommendations for lower-resource settings and potential collaborations with cloud service providers to offer access to necessary computational resources. Additionally, consider creating a simplified version of the benchmark that can be more easily adopted by researchers with limited resources.

4. **Ethical and Social Implications**:
   - **Current State**: The paper mentions safety checks for the Syn-IS dataset but does not delve deeply into the broader ethical and social implications.
   - **Improvement Suggestion**: Expand the discussion on the potential long-term impacts of deploying advanced OOD detection systems. Address issues such as the implications for privacy, security, and the potential for misuse. Discuss safeguards and ethical guidelines for the responsible use of these technologies in various applications.

5. **Future Directions**:
   - **Current State**: The paper acknowledges the need for further research but lacks specific suggestions.
   - **Improvement Suggestion**: Provide concrete recommendations for future research directions. This could include exploring new model architectures, developing specialized datasets for particular types of semantic and covariate shifts, and investigating cross-disciplinary approaches that combine OOD detection with other fields such as natural language processing and computer vision.

### Rewarding Upfront Acknowledgment
The authors should be commended for their upfront acknowledgment of the limitations and potential societal impacts of their work. This transparency is essential for fostering a responsible and ethical research community. By addressing these critical points and providing constructive feedback, the authors can further enhance the impact and relevance of their research.

### Summary
Overall, while the authors have made significant strides in addressing some limitations and societal impacts, there is room for a more comprehensive discussion and more detailed recommendations. Expanding on these areas will not only strengthen the paper but also provide valuable guidance for the research community moving forward.

**Opportunities For Improvement:**

### Limitations of "Rethinking the Evaluation of Out-of-Distribution Detection: A Sorites Paradox"

#### Significance of the Contribution
- **Scope of Benchmark:** While the IS-OOD benchmark addresses semantic and covariate shifts, it may not encompass all types of distribution shifts encountered in real-world applications. This limitation could affect its generalizability to broader OOD detection scenarios.
- **Dependence on CLIP Model:** The LAID method relies heavily on the alignment properties of the CLIP model. If these properties are not robust across different datasets or models, the effectiveness of the shift measurements might be compromised.

#### Relevance to the Broader Research Community
- **Generalization to Other Domains:** The benchmark and methods are primarily evaluated on image data. Their applicability and effectiveness in other domains, such as text or audio, remain uncertain and require further investigation.
- **Adoption by Community:** The adoption of the IS-OOD benchmark by the broader research community is crucial for its impact. There might be initial resistance or slow uptake as researchers need to adapt to this new benchmark and incorporate it into their evaluation frameworks.

#### Quality of the Research
- **Alignment Imperfections:** The alignment between the text and image feature spaces in the CLIP model is not perfect. This imperfection can introduce inaccuracies in the feature decomposition, potentially affecting the validity of the shift measurements and the resulting benchmark evaluations.
- **Generated Data Patterns:** The Syn-IS dataset, composed of generated images, may contain artifacts or patterns unique to the generation process. These unique features could influence the performance of OOD detection methods differently than real-world images, potentially limiting the generalizability of the findings.
- **Complexity of Evaluation:** The detailed methodology and multiple metrics used for evaluation, while thorough, may add complexity for researchers trying to replicate or build upon this work. Simplifying some aspects without losing rigor could make the research more accessible.

#### Ethical and Social Implications
- **Bias in Synthetic Data:** Although safety checks were conducted, there remains a risk of subtle biases in the synthetic data that could influence model evaluations. Ensuring that the synthetic data accurately represents the diversity and complexity of real-world data

**Relation To Prior Work:**

Yes

**Summary And Contributions:**

#### Introduction
The paper addresses the issue of Out-of-Distribution (OOD) detection, highlighting the challenges posed by marginal OOD samples that have close semantic content to In-Distribution (ID) samples. The authors identify this problem as a Sorites Paradox, where it becomes difficult to determine when a sample transitions from being ID to OOD.

#### Contributions
1. **Incremental Shift OOD (IS-OOD) Benchmark**: The authors construct a new benchmark that divides test samples into subsets with different semantic and covariate shift degrees relative to the ID dataset. This is achieved using a shift measuring method based on their proposed Language Aligned Image feature Decomposition (LAID).

2. **Synthetic Incremental Shift (Syn-IS) Dataset**: To complement the IS-OOD benchmark, the authors create a synthetic dataset containing high-quality generated images with diverse covariate contents. This dataset addresses the limited covariate variation in existing datasets.

3. **Language Aligned Image feature Decomposition (LAID)**: The LAID method is introduced to measure semantic and covariate shifts separately. This method leverages CLIP features to decompose and align text and image features, enabling accurate shift measurement.

4. **Evaluation and Insights**: The paper evaluates current OOD detection methods on the IS-OOD benchmark, uncovering several key insights:
   - Performance improves with increasing semantic shift.
   - Some methods, like GradNorm, rely less on semantic shifts.
   - Excessive covariate shifts can also be detected as OOD by some methods.

#### Main Findings
1. **Semantic Shift Sensitivity**: Most OOD detection methods perform better with larger semantic shifts. The benchmark highlights that the degree of semantic shifts significantly influences OOD detection capabilities.

2. **Covariate Shift Sensitivity**: While most methods are disturbed by covariate shifts, their primary factor for detection remains semantic shifts. However, in the synthetic dataset (Syn-IS), excessive covariate shifts are more likely to be considered OOD, revealing a different behavior from the real data.

3. **Evaluation Metrics**: The authors use metrics such as FPR@95, AUROC, and AUPR to evaluate the performance, and introduce additional metrics to study the changes in model performance across different shift levels.

#### Conclusion
The paper proposes a novel benchmark and synthetic dataset to better evaluate OOD detection methods, addressing the limitations of current benchmarks that use semantic labels to distinguish OOD samples. By focusing on the degree of shifts, the authors provide a more nuanced approach to OOD detection evaluation. They also highlight the need for future research to improve the alignment between text and image feature spaces to enhance the accuracy of shift measurements.

### Contributions Summary
- **Benchmark**: Incremental Shift OOD (IS-OOD) and Synthetic Incremental Shift (Syn-IS) datasets.
- **Method**: Language Aligned Image feature Decomposition (LAID) for measuring semantic and covariate shifts.
- **Evaluation**: Insights into OOD detection performance based on semantic and covariate shifts, with recommendations for future improvements.

This paper provides valuable resources and methodologies for advancing the research in OOD detection, addressing significant gaps in current benchmarks and proposing innovative solutions.

---

> ### Author Rebuttal · Authors · 2024-08-15
>
> **Limitation**
>
> **Bias and Fairness Analysis:**
>
> Good Point. We observed a certain correlation between semantic and covariate information in ImageNet-21K, as shown in Appendix C. This correlation could potentially lead to incorrect evaluations of OOD detection models that are sensitive to covariate shifts. By creating a more balanced dataset like Syn-IS, the bias caused by this correlation can be mitigated. More discussion will be added in the Appendix.
>
> **Gap Between Feature Spaces:**
>
> Thanks for the kindly suggestion. As we mentioned in the limitation section, this is indeed a valuable issue worth further investigating. Using features from more advanced vision-language models, such as BLIP or CoCa, could potentially further enhance the accuracy of our LAID method, which we will explore in the future. We will include a more detailed future work section on it in the Appendix.
>
> **Complexity and Accessibility:**
>
> Good suggestion. We will provide a simplified version of our benchmark in the future to facilitate its use by other researchers.
>
> **Ethical and Social Implications:**
>
> Thank you for your reminder. During the manual safety checks, we also examined the potential negative ethical and social implications of the data.
>
> Regarding the impacts on OOD detection systems, we believe that our benchmark can better assist models in improving both their semantic shift detection capabilities and their generalization to covariate shifts, thereby ensuring their robustness across various scenarios.
>
> We hope the response has addressed your concerns. If our understanding is not accurate, we would greatly appreciate any further clarification you could provide.
>
> **Future Directions:**
>
> Thank you for your suggestion. We give a more concrete future work and will include it in the Appendix.
>
> “In terms of improving the benchmark, a promising direction for future research is to explore the use of more advanced vision-language models, such as BLIP or CoCa, to reduce the gap between text and image feature space, thereby enhancing the accuracy of feature decomposition in the LAID method. Additionally, separate investigations of particular covariate shifts are valuable, such as specifically studying the impact of changes in artistic styles on OOD detection model performance. We also strongly encourage subsequent researchers to use our benchmark for further evaluation of models in other fields, such as assessing whether the understanding of image semantic information by the current large-scale vision-language models, such GPT-4, is affected by image covariate shifts.”

---

> ### Author Rebuttal · Authors · 2024-08-15
>
> **Correctness**
>
> **Detailed Justification of Claims:**
>
> Thanks for the comments. Some of the most relevant existing benchmarks and the comparative analyses are shown below, which can be also found in the Appendix A.
>
> “Compared to previous works that remove the noisy marginal ID samples from the OOD dataset [1-3], our approach does not involve manual annotation and thus introduces less subjective bias. Compared to a work that considers the impact of covariate contents [4], IS-OOD analyzes the covariate shifts in more detailed levels and considers them in conjunction with semantic shifts, which allows for a more comprehensive evaluation of how covariate contents affect the OOD detection model.
>
> An interesting work also proposes to divide test data into different subsets for evaluation [5], in which ImageNet-21K is divided based on the OOD detection difficulty. However, they do not consider the impact of the covariate shifts in the benchmark. Besides, since the "detection difficulty" is derived from models trained only on the training set, its accuracy on the test data cannot be guaranteed.”
>
> **Dataset Annotation Details:**
>
> Thanks for the suggestion.
>
> As shown in the annotating process of ImageNet-21K subsets in Appendix C, the annotator is essentially a combination of the trained transformation matrix and the CLIP feature extractor. As described in section 2.1, in each training iteration, we select a batch of standard text along with their corresponding semantic shift text and covariate shift text. We use the CLIP feature extractor and the transformation matrix to obtain the semantic and covariate features of these texts. We then calculate the loss to optimize the parameters of the transformation matrix.
>
> The detailed codes for training the annotator are going to be open-sourced.
>
> **Error Analysis Expansion:**
>
> Thanks for the good suggestion. We have included examples of prediction errors by the OOD detection model in the attached **PDF**. As observed, the model does indeed assign high confidence (i.e., low OOD scores) to some marginal OOD samples. We will include a section in the Appendix to show and analyze these specific examples.
>
> **Evaluation Metrics:**
>
> Thanks for the comments. We employed FPR@95, AUROC, and AUPR in our experiments to maintain consistency with previous works, in which AUPR includes measures of both recall and precision. Additionally, to assess the performance of OOD detection models across different shift levels, we introduced two new metrics for evaluation.
>
> We first use the Pearson correlation coefficient to evaluate the relationship between the model‘s performance and the shift levels.
>
> $correlation = \frac{\sum_1^n(x_i-\bar x)(i-\frac{n+1}{2})}{\sqrt{\sum_1^n(x_i-\bar x)^2\sum_1^n(i-\frac{n+1}{2})^2}}$
>
> To further investigate the extent of changes in model performance, we define a model's sensitivity to the corresponding shifts.
>
> $sensitivity = \left| \frac{\sum_1^n(x_i-\bar x)(i-\frac{n+1}{2})}{\sum_1^n(i-\frac{n+1}{2})^2} \right|$
>
> Could you please provide more details about what you mean by " breakdowns by question type and document type "? This will help me better understand your suggestion.
>
>
>
> [1] J. Yang, H. Wang, L. Feng, X. Yan, H. Zheng, W. Zhang, and Z. Liu, “Semantically coherent out-of-distribution detection,” in IEEE Int. Conf. Comput. Vis. (ICCV), 2021, pp. 8301–8309.
>
> [2] J. Bitterwolf, M. Mueller, and M. Hein, “In or out? fixing imagenet out-of-distribution detection evaluation,” Int. Conf. Mach. Learn. (ICML), 2023.
>
> [3] W. Yang, B. Zhang, and O. Russakovsky, “Imagenet-ood: Deciphering modern out-of-distribution detection algorithms,” Int. Conf. Learn. Represent. (ICLR), 2024.
>
> [4] J. Yang, K. Zhou, and Z. Liu, “Full-spectrum out-of-distribution detection,” Int. J. Comput. Vis. (IJCV), vol. 131, no. 10, pp. 2607–2622, 2023.
>
> [5] I. Galil, M. Dabbah, and R. El-Yaniv, “A framework for benchmarking class-out-of-distribution detection and its application to imagenet,” Int. Conf. Learn. Represent. (ICLR), 2023.

---

### Decision · Program_Chairs · 2024-09-26

**Decision:**

Accept (Poster)

**Comment:**

The paper addresses the issue of Out-of-Distribution detection, highlighting the challenges posed by marginal OOD samples that have close semantic content to In-Distribution samples. After repeated communication between the author and the reviewers, the relevant doubts and uncertainties were resolved. Ultimately, based on various factors, including innovation and rigor, I have determined that this paper can be accepted.